# Poor sleep and shift work associate with increased blood pressure and inflammation in UK Biobank participants

Monica Kanki[1,2,9], Artika P. Nath[3,9], Ruidong Xiang[3,9], Stephanie Yiallourou[4], Peter J. Fuller [5], Timothy J. Cole [6], Rodrigo Cánovas[3,7] & Morag J. Young [1,8] ✉

Disrupted circadian rhythms have been linked to an increased risk of hypertension and cardiovascular disease. However, many studies show inconsistent findings and are not sufficiently powered for targeted subgroup analyses. Using the UK Biobank cohort, we evaluate the association between circadian rhythm-disrupting behaviours, blood pressure (SBP, DBP) and inflammatory markers in >350,000 adults with European white British ancestry. The independent U-shaped relationship between sleep length and SBP/DBP is most prominent with a low inflammatory status. Poor sleep quality and permanent night shift work are also positively associated with SBP/DBP. Although fully adjusting for BMI in the linear regression model attenuated effect sizes, these associations remain significant. Two-sample Mendelian Randomisation (MR) analyses support a potential causal effect of long sleep, short sleep, chronotype, daytime napping and sleep duration on SBP/DBP. Thus, in the current study, we present a positive association between circadian rhythm-disrupting behaviours and SBP/DBP regulation in males and females that is largely independent of age.

High blood pressure (BP) is a major risk factor for poor cardiovascular health outcomes, which are the lead cause of mortality worldwide[1,2]. Increased exposure to blue light, eating, and activity during the biological rest period can rapidly disrupt internal circadian rhythms and our physiological adaptation to light-dark cycles, which compromise organ function[3–5]. Therefore, improving our understanding of the impact of non-traditional lifestyle factors and work practices is crucial for implementing appropriate preventative measures to minimise the risk of hypertension[6,7].

Metanalyses indicate that both short (<5 h) and long sleep (>8 h) are associated with an increased risk of hypertension and other health complications; however, inconsistent findings linking sleep length with risk of hypertension have been reported across study populations[8]. The Coronary Artery Risk Development in Young Adults (CARDIA) study showed short sleep predicted a 37% increase in the risk of hypertension for every hour reduction in sleep length following adjustment for age, sex and ethnicity[9]. However, sleep length alone may not suffice as an adequate measure of circadian burden and it is only when combined with other factors resulting in sleep disturbances that sleep behaviour can provide an overall impression of sleep health[10,11]. Furthermore, a previous study using UK Biobank cohort data revealed that shift workers who slept <6 h took more BP-lowering medication than non-

[1]Cardiovascular Endocrinology Laboratory, Baker Heart and Diabetes Institute, Melbourne, VIC, Australia. [2]Department of Medicine (Alfred Health), Central Clinical School, Monash University, Clayton, VIC, Australia. [3]Cambridge-Baker Systems Genomics Initiative, Baker Heart and Diabetes Institute, Melbourne, VIC, Australia. [4]Turner Institute for Brain and Mental Health, Department of Central Clinical School, Monash University, Clayton, VIC, Australia. [5]Centre of Endocrinology and Metabolism, Hudson Institute of Medical Research, Clayton, VIC, Australia. [6]Department of Biochemistry and Molecular Biology, Monash University, Clayton, VIC, Australia. [7]Health and Biosecurity, Australian e-Health Research Centre, CSIRO, Melbourne, VIC, Australia. [8]Department of Cardiometabolic Health, University of Melbourne, Melbourne, VIC, Australia. [9]These authors contributed equally: Monica Kanki, Artika P. Nath, Ruidong Xiang. ✉e-mail: morag.young@baker.edu.au

shift workers[12]. Similarly, systematic reviews and meta-analyses do not present consistent effects of shift work on BP and cardiovascular disease, suggesting insufficient power in these studies[13,14].

Chronic low-grade inflammation is also an important contributor to the pathophysiology of hypertension[15]. Biomarkers of systemic inflammation, such as plasma C-reactive protein (CRP) levels and white blood cell counts, are associated with high BP and mortality related to cardiovascular diseases[16,17]. Furthermore, inflammatory responses are strongly gated by the circadian clock and glucocorticoid levels, both of which can become dysregulated in settings of circadian misalignment[18]. While shift work, reduced sleep and an evening chronotype, are independently associated with higher levels of circulating inflammatory markers, whether inflammatory factors are linked to associations between circadian rhythm-disrupting behaviours and BP has not been evaluated[19,20]. Similarly, previous studies have not evaluated such associations when data are stratified for BMI, sex, age or inflammatory status, and even fewer have evaluated the impact of shift work on blood pressure in a cohort of this size.

The UK Biobank is a unique and large-scale biomedical research resource containing in-depth genetic and health information from more than half a million volunteers[12,21]. This provides an opportunity to investigate associations between sleep length, sleep quality and shift work schedules with BP and their interaction with inflammation while accounting for a range of standard and non-standard confounders. While several recent studies provide support for a relationships between sleep phenotypes and cardiovascular diseases, these studies did not undertake detailed stratification of the data, or focused on one sleep trait only or investigated genetically predicted sleep duration[22-25]. Therefore, in the current study, we aimed to determine whether the positive association between circadian rhythm-disrupting behaviours including multiple sleep phenotypes and BP could be linked to systemic markers of inflammation. Moreover, we also performed two-sample linear and non-linear Mendelian Randomisation (MR) analyses using the genome-wide association study (GWAS) from the UK Biobank cohort and International Consortium of BP Genome-Wide Association Studies (ICBP) (~1 million people) to validate our findings and evaluate the causal effect of indicators of multiple sleep-related traits on BP. Our analyses reveal relationships between sleep phenotypes and BP, which may further the understanding of circadian patterns of sleep behaviour and blood pressure.

## Results

### Characteristics of the study population
A total of 426,308 UK Biobank participants (54% females), aged 37–73 years old (mean ± SD: 57 ± 8 years) with an average BMI of 27.3 ± 4.6 kg/m[2], had an overall mean SBP and DBP of 141 ± 20 and 84 ± 11 mmHg, respectively. Forty percent of participants reported sleeping an average of 7 hrs/day and 65% were assigned a moderate sleep quality score (Table 1). Mean SBP, DBP and BMI were lowest in individuals who reported sleeping an average of 7 hrs/day or in those assigned a healthy sleep score. Thus, these sub-groups were the nominated reference categories for the following linear regression analyses related to each sleep length and sleep quality, respectively. Participants who slept ≤5 hrs/day (4.8%) or ≥9 hrs/day (7.0%) had higher BP, were more likely to be diagnosed with hypertension and to use BP-lowering medications than those sleeping an average of 6, 7 or 8 hrs/day. Similarly, participants assigned a poor (12.0%) and moderate (64.9%) sleep quality score had higher BP, were older, more likely to be hypertensive and take BP-lowering medications compared to those who were assigned a healthy sleep quality score.

### Short/long sleep lengths and unhealthy sleep qualities are associated with BP
All sleep length categories were positively associated with SBP and DBP compared to 7 hrs/day of sleep in the baseline model ($p < 0.05$;

Fig. 1). The β-estimate detected was greatest for very short (≤5 hrs/day), followed by very long sleep lengths (≥9 hrs/day), versus SBP and DBP, suggesting a U-shaped relationship with BP (SBP β = 1.13 and 0.94, DBP β = 1.03 and 0.97 for ≤5 and ≥9 hrs/day, respectively; $p$-value < 0.05) (Fig. 1, Supplementary Table S1). Poor and moderate sleep qualities were also positively associated with a BP versus healthy sleep scores in the baseline model (β = 0.81 and 1.66 for SBP and 0.90 and 1.78 for DBP, respectively, $p < 0.05$) (Fig. 1, Supplementary Table S1).

Adjusting for BMI reduced the β-estimate for the associations between sleep length and BP such that only the association between long sleep lengths (8 h and ≥9 h/day) and SBP/DBP remained statistically significant ($p < 0.05$ in the BMI adj model; Fig. 1, Supplementary Table S1). Similarly, adjusting for BMI attenuated the effect size for the association between sleep quality and both SBP and DBP; however, the association remained significant for poor and moderate sleep qualities and DBP ($p < 0.05$). Removing participants on antihypertensive medications produced similar results in the 'Antihypertensive (AH) free', and 'AH free BMI adj' models (Supplementary Table S1).

The impact of BMI, sex or age on the identified associations was evaluated by performing multiple linear regression analyses after stratifying datasets according to each covariate. Sleep lengths of ≤5, 8 h and ≥9 hrs/day remained positively associated with DBP in all three BMI categories ($p < 0.05$ in the baseline model; Supplementary Fig. S1 and Table S2). However, the U-shaped association between sleep length and SBP was lost in participants with a BMI of 25- < 30 or ≥30 kg/m[2]. Furthermore, while a significant U-shape association between sleep length and BP remained in males with a BMI > 25 kg/m[2], the positive association between short sleep lengths and SBP/DBP were lost in males with a BMI ≤ 25 kg/m[2] (Supplementary Table S3). Similarly, only 8 and ≥9 hrs/day remained positively associated with SBP/DBP in females with a BMI of ≤25 or >25 kg/m[2] ($p < 0.05$). When data were stratified according to sex or age ( ≤ 50 yr versus >50 yr), the U-shaped association between sleep length and SBP/DBP remained ($p < 0.05$ in baseline; Supplementary Figs. S2, S3 and Table S4). However, the positive association between ≤5 hrs/day and SBP was lost in males and females, and the association between ≤5 h/day and SBP/DBP was lost in individuals ≤50 years old when the model was further adjusted for BMI ($p < 0.05$ in the BMI adj model).

Stratification of the sleep quality dataset by BMI showed that poor and moderate sleep qualities were positively associated with DBP in all BMI sub-groups ($p < 0.05$ in the baseline model; Supplementary Fig. S4 and Table S5). However, only moderate sleep quality remained positively associated with SBP in participants with a BMI of ≥30 kg/m[2] group while all other associations between sleep quality and SBP were lost. Moreover, a larger effect size (i.e., β-estimate) was detected for the significant association between each poor and moderate sleep quality with SBP/DBP in females versus males when data were stratified by sex ($p < 0.05$ in the baseline model; Supplementary Fig. S5 and Table S6). Furthermore, the positive association between sleep quality and BP was negated in only males following correction for BMI in all cases except for moderate sleep quality and DBP ($p < 0.05$ in the BMI adj model; Supplementary Fig. S5 and Table S6). Stratifying by age revealed that only the associations between poor and moderate sleep qualities with DBP remained significant following adjustment for BMI ($p < 0.05$ in baseline and BMI adj models; Supplementary Fig. S6 and Table S6).

### Permanent night shift work is positively associated with DBP
At the time of recruitment, 15.3% of currently employed individuals reported shift work schedules with more males than females in each shift work sub-category (Table 1). Participants who undertook mixed, night or permanent night shift work were younger, had higher BP and a

**Table 1 | General characteristics of participants based on their sleep length, sleep quality score and shift work status**

| Sleep length | | | | | |
|---|---|---|---|---|---|
| | ≤5 h | 6 h | 7 h | 8 h | ≥9 h |
| Number of participants | 189751 (5.2%) | 740321 (20.1%) | 1567331 (42.5%) | 1159621 (31.1%) | 27347 (0.7%) |
| Age (years) mean ± SD | 57.38 ± 7.68 | 56.63 ± 7.77 | 56.11 ± 8.01 | 57.35 ± 8.13 | 59.20 ± 7.81 |
| BMI (kg/m²) mean ± SD | 28.27 ± 5.15 | 27.63 ± 4.74 | 27.00 ± 4.43 | 27.04 ± 4.42 | 27.83 ± 4.82 |
| SBP (mmHg) mean ± SD | 142.21 ± 20.15 | 140.81 ± 19.79 | 139.90 ± 19.93 | 141.26 ± 20.55 | 143.92 ± 20.82 |
| DBP (mmHg) mean ± SD | 84.73 ± 10.98 | 84.15 ± 10.90 | 83.58 ± 10.84 | 83.89 ± 10.95 | 85.05 ± 10.97 |
| % Female | 10,540 (56%) | 38,152 (52%) | 81,279 (52%) | 64,439 (56%) | 14,903 (54%) |
| Antihypertensive medication | 2106 (11%) | 5760 (7.8%) | 9405 (6.0%) | 8341 (7.2%) | 3119 (11%) |
| History of hypertension | 3925 (21%) | 12,677 (17%) | 23,038 (15%) | 19,610 (17%) | 6403 (23%) |
| *Sleep quality score* | | | | | |
| | *4 to 5* | *2 to 3* | | *0 to 1* | |
| Number of participants | 59787 (23.1%) | 168431 (64.9%) | | 31153 (12.0%) | |
| Age (years) mean ± SD | 55.10 ± 8.25 | 56.59 ± 8.01 | | 57.81 ± 7.76 | |
| BMI (kg/m²) mean ± SD | 26.30 ± 4.16 | 27.20 ± 4.47 | | 28.60 ± 4.83 | |
| SBP (mmHg) mean ± SD | 138.03/82.58 ± 19.84 | 140.48 ± 20.06 | | 143.15 ± 20.05 | |
| DBP (mmHg) mean ± SD | 82.58 ± 10.68 | 83.91 ± 10.90 | | 85.45 ± 10.98 | |
| % Female | 32471 (54%) | 88389 (52%) | | 14059 (44%) | |
| Antihypertensive medication | 9134 (12%) | 36032 (17%) | | 9776 (23%) | |
| History of hypertension | 3587 (4.9%) | 15708 (7.3%) | | 4557 (11%) | |
| *Shift work* | | | | | |
| | *No shift work* | *Day shift* | *Mixed shift* | *Night shift* | *Permanent night shift* |
| Number of participants | 183518 (83.7%) | 17480 (8.0%) | 10150 (4.6%) | 3167 (1.4%) | 4716 (2.2%) |
| Age (years) mean ± SD | 52.55 ± 6.92 | 52.27 ± 6.89 | 51.01 ± 6.72 | 50.92 ± 6.68 | 51.23 ± 6.68 |
| BMI (kg/m²) mean ± SD | 26.92 ± 4.44 | 27.55 ± 4.72 | 28.00 ± 4.62 | 28.09 ± 4.69 | 28.28 ± 4.51 |
| SBP (mmHg) mean ± SD | 136.30 ± 18.87 | 136.44 ± 18.79 | 137.26 ± 18.41 | 136.87 ± 17.86 | 138.29 ± 18.37 |
| DBP (mmHg) mean ± SD | 83.16 ± 10.89 | 83.43 ± 10.94 | 84.32 ± 11.01 | 84.14 ± 10.72 | 84.81 ± 10.92 |
| % Female | 77733 (51%) | 6735 (49%) | 2715 (34%) | 772 (32%) | 1102 (31%) |
| Antihypertensive medication | 19698 (11%) | 2024 (12%) | 1171 (12%) | 366 (12%) | 559 (12%) |
| History of hypertension | 7055 (3.8%) | 830 (4.7%) | 451 (4.4%) | 148 (4.7%) | 223 (4.7%) |

higher BMI compared to non-shift workers. Diagnosis of hypertension and use of antihypertensives was similar across all types of shift work.

Mixed and permanent night shift work were positively associated with SBP and DBP versus non-shift workers in the baseline model ($p < 0.05$; Fig. 2 and Supplementary Table S1). Adjusting for BMI substantially attenuated the positive association between work schedules and BP ($p > 0.05$ in the BMI adj model; Fig. 2 and Supplementary Table S1). In contrast, when data were stratified by BMI, only permanent night shift was significantly associated with DBP in individuals with a BMI > 30 kg/m² ($p < 0.05$ in the baseline model; Supplementary Fig. S7 and Table S7). Interestingly, only day shift work was positively associated with SBP in males with a BMI of ≤25 kg/m², while permanent mixed and permanent night shift were positively associated with DBP in females with a BMI > 25 kg/m² ($p < 0.05$ in the baseline model and Supplementary Table S8).

When datasets were stratified by sex, mixed and permanent night shift work were only positively associated with SBP and DBP in males ($p < 0.05$ in the baseline model; Supplementary Fig. S8 and Table S9). Adjusting for BMI negated these associations in males except for the association between permanent night shift and DBP ($p < 0.05$ in the BMI adj model; Supplementary Fig. S8 and Table S9). When data were stratified by age, mixed and permanent night shift work were positively associated with DBP, but not SBP, in both age groups ($p < 0.05$ in the baseline model; Supplementary Fig. S9 and Table S9). Adjusting the model for BMI attenuated all positive associations and resulted in a weak negative association between each day shift, mixed and night shift work with SBP and DBP ($p < 0.05$ in the BMI adj model; Supplementary Fig. S9 and Table S9).

## Shift work, sleep length and BP

The combined effect of shift work and sleep length was assessed by an interaction analysis whereby all effect sizes for the associations with SBP/DBP are presented compared to non-shift workers sleeping 7 hrs/day. In day shift workers very short and very long sleep were also positively associated with modestly DBP ($p < 0.05$ in the baseline model; Fig. 3 and Supplementary Table S10). However, the β-estimate is greatest for the associations between permanent night shift workers sleeping ≤5 or 6 h/day and SBP/DBP (SBP β = 2.57 and 1.18, DBP β = 2.29 and 1.50 for ≤5 and 6 hrs/day, respectively; $p$-value < 0.05) (Supplementary Table S10). Further adjusting for BMI in the baseline model negated all positive associations ($p > 0.05$ in the BMI adj model).

## Sleep length and BP associations depend on inflammatory status

Given that inflammation is associated with increased BP, we investigated the effect of adjusting for levels of CRP alone or in combination with BMI on the associations between sleep length, sleep quality, shift work and SBP/DBP[15]. Our data showed that further adjusting for CRP or CRP plus BMI had no effect on primary outcomes of the BMI adj model, which reflects the strong correlation between BMI and CRP in the correlation plot (Supplementary Table S11 and S12). Poor sleep health (<5 h and >9 h, and poor sleep quality) and night shift work were positively correlated with CRP levels, and leukocyte and neutrophil counts (Supplementary Table S11).

Regression analyses were repeated in datasets stratified according to low, medium and high levels of CRP, and terciles of circulating lymphocyte, monocyte, and neutrophil cell counts. The U-shape

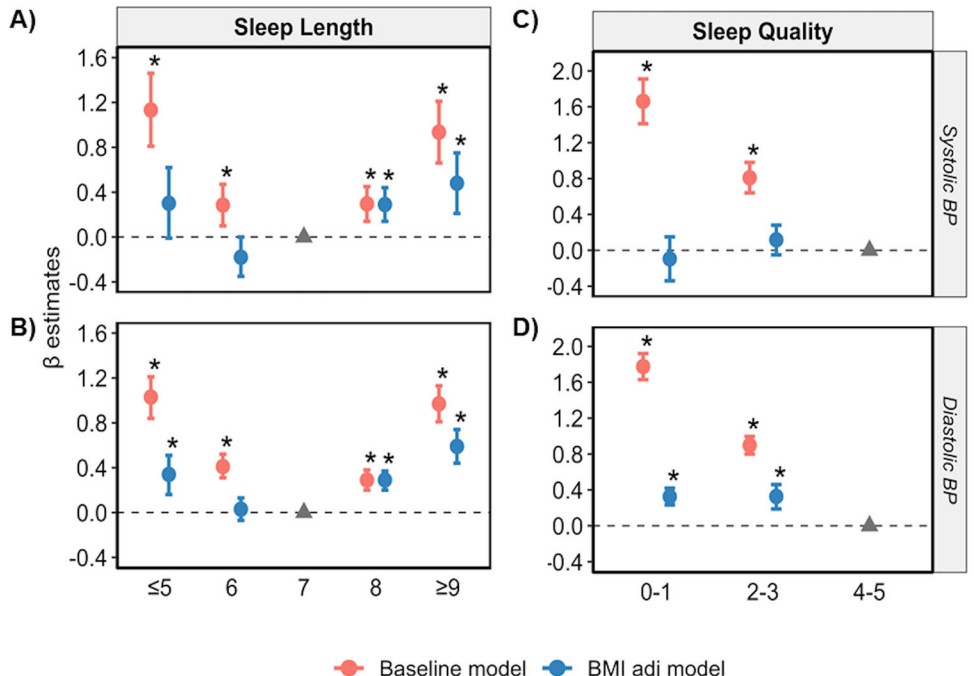

**Fig. 1 | A U-shaped association exists between sleep length and SBP/DBP is present even when adjusted for BMI.** Relationship between sleep length (**A**, **B**) and sleep quality (**C**, **D**) versus SBP and DBP, respectively, with (baseline model, red) and without adjustment for BMI (BMI adj model, blue). Data are expressed as β- estimate ± 95% CI, which are presented as the centre circle and corresponding error bars, respectively. P-values are estimated using multivariate logistic regression. *p < 0.05 versus sleep length of 7 h or a sleep quality score of 4–5, respectively, denoted by the solid grey triangle. Please see also Supplementary Table S1.

association between sleep length and SBP/DBP was only apparent in low and medium terciles in the baseline model (Figs. 4–7, Supplementary Tables S13–S14). Poor and moderate sleep qualities were positively associated with SBP and DBP regardless of inflammatory marker levels (*p* < 0.05 in the baseline model; Supplementary Tables S15–S16). However, the positive association detected for poor and moderate sleep quality with DBP (but not SBP) in low and medium levels of all four inflammatory markers remained significant even when adjusting for BMI in the model (*p* < 0.05 in the baseline and BMI adj models). The positive association between shift work and BP in terciles for inflammatory markers were lost when adjusted for BMI (*p* > 0.05 in the BMI adj model; Supplementary Tables S17–S18).

**Short and long sleep lengths, chronotype daytime sleepiness and daytime napping are causally linked to BP**

To estimate the causal effects of sleep length or sleep quality traits on SBP and DBP, we conducted two-sample MR analyses using published GWAS summary statistics[26–29]. We used Generalized Summary-data-based MR (GSMR)[30] to analyse GWAS summary data and then used MR-Egger and weighted median sensitivity analyses implemented in Yavorska et al.[31] to verify results from GSMR. Based on the agreement between the three methods, we found significant causal associations between SBP/DBP and long or short sleep durations, as well as chronotype and daytime napping (Fig. 8). Although some causal associations were not consistent between the GSMR, weighted median and MR-Egger MR analyses, daytime napping was identified as causally associated with SBP and DBP using all three approaches (Table 2).

As sleep duration displayed non-linear effects in the multiple linear regression analyses, we further explored causal associations using non-linear MR analyses specifically for sleep duration and BP. We observed significant non-linear causal associations between sleep duration and SBP/DBP in the PolyMR[32] and SUMnlmr[33] methods (Supplementary Fig. S10 and S11). Although this significant non-linear causal association was detected, we also note that the association between sleep duration and DBP appeared rather linear, suggesting a

linear causal association is favoured between sleep duration and BP (Supplementary Fig. S10).

## Discussion

We used the UK Biobank cohort, the largest study of its kind with over ~350,000 participants, to demonstrate independent associations between sleep length, sleep quality and shift work with BP. Our data therefore demonstrate positive associations between short/long sleep lengths, poor and moderate sleep qualities and permanent-night shift work and BP. Following adjustment for BMI, a strong positive association remained between sleep length, permanent night shift work, and DBP, whereas the association for SBP was attenuated. Similarly, while individuals with poor sleep quality associations appeared mostly attenuated after adjusting for BMI in the model, particularly for SBP; however, appeared to a greater impact on BP in females versus males. Two-sample MR analysis demonstrated that both short and long sleep duration, chronotype and daytime napping were causally linked to BP using linear MR. In non-linear MR analyses, a causal association was detected for sleep duration and SBP/DBP. Taken together our data support an independent link between circadian rhythm-disrupting behaviours and BP, supporting the importance of sleep health and lifestyle factors such as work schedules for BP management.

Both short and long sleep duration were positively associated with BP compared to 7 h, which is consistent with previous studies that have demonstrated a U-shape relationship between sleep length and risks of hypertension and other cardiovascular disease risk factors[3,4]. Sex, age, BMI and inflammation are some of the key covariates that have been linked to the adverse impact of circadian dysregulation on cardiovascular disease[34]. For example, the prevalence of cardiovascular disease and risk factors including hypertension dramatically increases in post-menopausal to match that of men, and survivorship following time of day-dependent myocardial infarction also shows sexual dimorphism[35,36]. Furthermore, the robustness of biological circadian rhythms exhibits age-dependent declines and can be blunted with higher BMI and pro-inflammatory markers[37–39]. The size of the UK

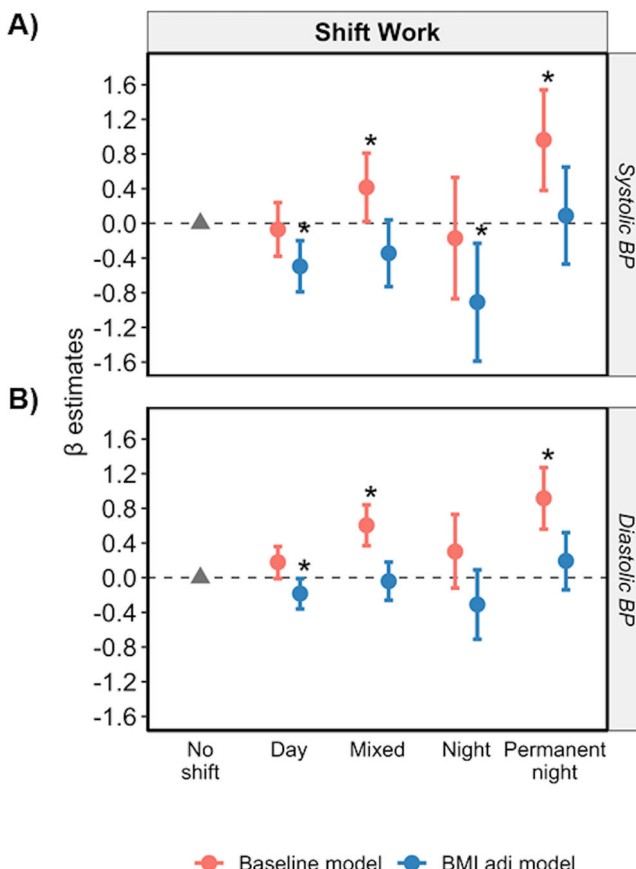

**Fig. 2 | Mixed shift and permanent night shift work are positively associated with SBP and DBP.** Relationship between shift work schedule versus SBP (**A**) and DBP (**B**), respectively, with (baseline model, red) and without adjustment for BMI (BMI adj model, blue). Data are expressed as β-estimate ± 95% CI, which are presented as the centre circle and corresponding error bars, respectively. *P*-values are estimated using multivariate logistic regression. *$p < 0.05$ versus no shift work group denoted by the solid grey triangle. Please see also Supplementary Table S1.

Biobank dataset enabled detailed subgroup analyses, which further revealed that this positive association between non-standard sleep length (<7 h<) and BP was largely independent of some established demographics including sex, age and BMI. Interestingly, we found that associations between sleep length, sleep quality and permanent night shift work with SBP were largely lost when adjusting for BMI in the regression model or separately stratifying datasets by BMI sub-groups. In contrast, associations between sleep length and sleep quality with DBP remained positive in the BMI adj model, suggesting poor sleep health or shift work potentially influences underlying tissue mechanisms to adversely increase DBP that could be independent of BMI[40–42].

In the present study, we identified a causal association between long and short sleep duration with SBP and DBP, suggesting an optimal amount of sleep is required to support a healthy level of BP. Similarly, sleep traits including chronotype, daytime sleepiness and daytime napping were also causally linked to SBP and DBP in linear MR analyses. These data are consistent with previous studies that have found that genetically predicted sleep traits (chronotype, daytime sleepiness, long/short sleep lengths, insomnia) were largely causally linked to an elevated risks of hypertension and heart failure[22,43]. Sleep traits have also been causally associated with cardiometabolic risk factors such as BMI, which may partly explain why positive associations between poor and moderate sleep qualities and BP were dramatically modulated in models further adjusted for BMI in our regression analyses[44,45].

Interestingly, we also observed some tendency of non-linear causal relationships between sleep duration and SBP and DBP. This suggests that sleep length could be a potential non-linear causal factor for increased BP; however, a linear relationship is likely when addressing the causal association for optimal sleep length (i.e., long or short sleep lengths separately). Recently, Liang et al. (2023) and Ai et al. (2021) proposed an L-shaped causal relationship between genetically predicted sleep length and risk of metabolic syndrome and select cardiovascular diseases, including coronary artery disease, arterial hypertension, chronic ischemic heart disease, myocardial infarction, and metabolic syndrome, suggesting that short sleep length is more profoundly linked to adverse outcomes[24,46]. However, non-linear relationships are difficult to interpret and replicate. While observational data and regression analyses present a positive association between short and long sleep duration and cardiovascular disease risk, several studies have not found a consistent causal association between genetically predicted long sleep length and risk of cardiovascular diseases including myocardial infarction, heart failure, coronary artery disease and stroke using MR-based approaches[47]. Therefore, we caution that the findings from our non-linear MR will also require confirmatory evidence from future studies. However, taken together, these data provide strong support for our initial association studies that revealed a significant association between circadian disrupting traits and BP, and provide additional evidence for the importance of sleep length for a key risk factor for cardiovascular disease.

Interestingly, this impact of BMI on SBP is consistent with other reports that propose systolic hypertension as a potent determinant and predictor of cardiovascular disease-related outcomes[48]. SBP and DBP are regulated by multiple factors including sympathetic activity, heart rate, vascular elasticity and blood flow where a slower heart rate, higher arterial elasticity and arteriolar resistance increase DBP[49]. SBP and DBP differ in that SBP increases with age while DBP is often lower with ageing. Arteriolar resistance is a key component of total peripheral resistance (TPR) and is regulated by numerous neurohormonal factors controlling sympathetic tone and vasoactive factors such as endothelin 1, nitric oxide production, angiotensin II/aldosterone and inflammatory mediators[49]. While direct measures of circadian misalignment were not assessed in this study, circadian disruption has been previously linked to changes in each of these humoral factors where even short-term simulated shift work modulates adrenal steroid release and bone marrow haematopoiesis[50]. This could offer insights into potential modifiable mechanisms that are targeted to reduce the impact of circadian disruption on BP in otherwise healthy individuals.

Shift workers are a particularly vulnerable group that experience compromised sleep length and poor sleep quality, and increase risks of hypertension and diabetes even years after retirement[51]. Permanent night shift work had a positive association with SBP and DBP that was consistent across the stratified and non-stratified datasets. Our findings complement previous reports and meta-analyses that identified an association between shift work and elevated BP, which we showed was strongest for permanent night shift work versus rotating shift work[52]. However, the association between shift work schedules other than permanent night shift and SBP or DBP in this study or others are not consistent and thus, are not well known[52,53]. A unique approach in our study design was to capture the adverse impact of different schedules of shift work on both SBP and DBP amongst mixed professions with varying sedentary levels. A limitation is that shift work schedules were assigned based on self-reported data, which made identifying differences between more nuanced shift work types is challenging. This could explain why associations between mixed or night shift work groups, BP and inflammation contrasted with previous cohort studies[54–56]. Combined short sleep and night shift work was most associated with SBP and DBP compared to all other sleep length categories. This suggests perhaps the previously identified adverse

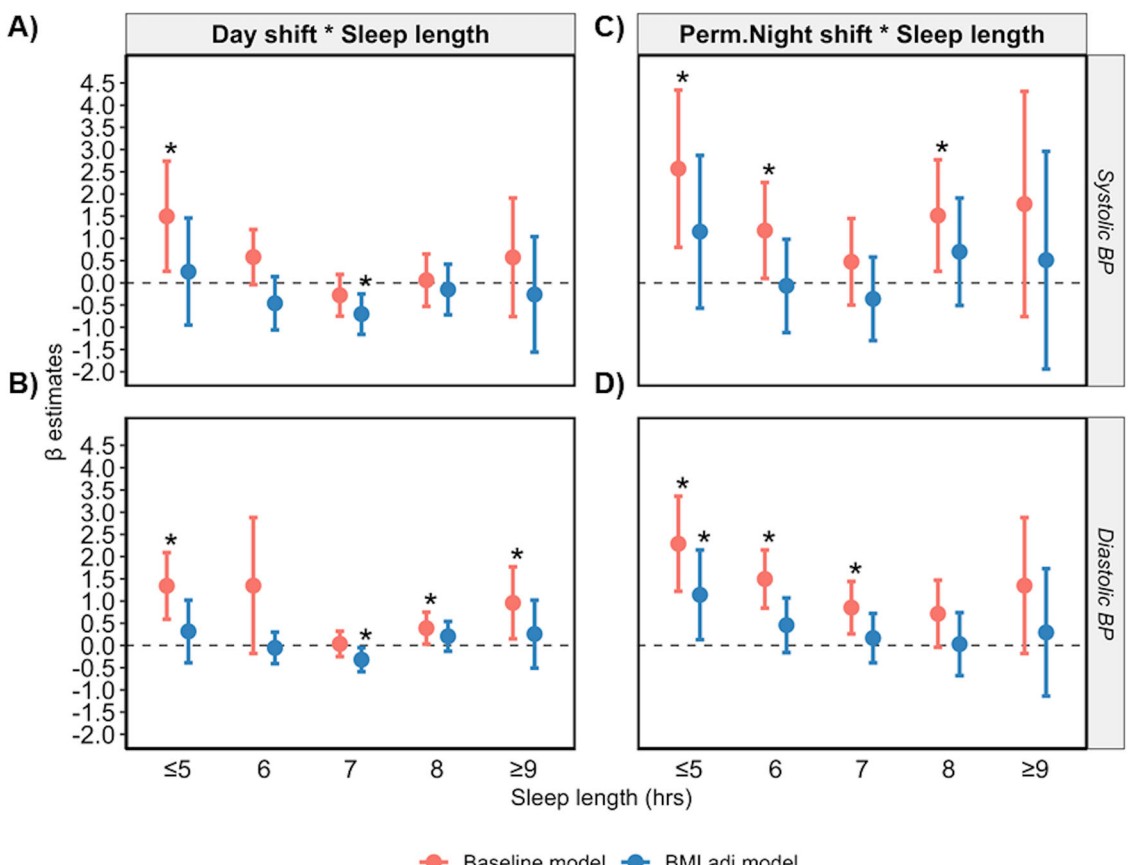

**Fig. 3 | A U-shaped relationship between sleep length and BP persists in day shift workers and permanent night shift workers.** Relationship between sleep length versus SBP and DBP in day shift (**A**, **B**) and permanent night shift workers (**C**, **D**) with (baseline model, red) and without adjustment for BMI (BMI adj model, blue). Data are expressed as β-estimate ± 95% CI, which are presented as the centre circle and corresponding error bars, respectively. *P*-values are estimated using multivariate logistic regression. $^*p < 0.05$ versus are non-shift workers who reported an average of is 7 h sleep per day denoted by the solid grey triangle. Please see also Supplementary Table S10A and S10B.

effect of long sleep duration is masked by the impact of altered sleep wake cycles on BP among shift workers.

This study addresses the link between inflammation, BP and circadian rhythm-disrupting behaviours in the UK Biobank cohort. We provide data supporting a loss of the this U-shaped relationship between sleep length and BP in individuals with a high inflammatory status. CRP levels, white blood cell counts and proinflammatory cytokines are recognised biomarkers for predicting risks of hypertension, cardiovascular and cardiometabolic conditions in large population-based studies[57,58]. Given the links between elevated systemic markers of inflammation and risks of hypertension have been described, we propose that a heightened inflammatory status masks the adverse effects of short or long sleep lengths on BP[15]. While our findings concur with other smaller cohort studies that link heightened levels of inflammatory markers with one or more forms of circadian misalignment or sleep loss[19,59], we provide data suggesting the positive association between SBP and sleep length was strongest in participants with low markers of inflammation. Previous studies using UK Biobank data have reported an association between changes in day length with cardiovascular disease-related mortality, BP, BMI and inflammation or white blood cell populations, with SBP, DBP and pulse pressure; however, these do not consider the effect of circadian rhythm-disrupting behaviours and work schedules[17,60]. Shift work induced circadian misalignment has also been linked to elevated serum CRP levels, BP, interleukin-6 and tumour necrosis factor-α and increased cardiovascular disease risk[61,62]. Taken together these findings support investigating independent associations

between circadian strain (or misalignment), circulating markers of inflammation and BP.

Our approach and analyses are limited by several factors. Analyses were restricted to participants of European White ethnicity (>94% of the UK Biobank database) as all other ethnic groups are poorly represented in this dataset. To avoid under-powered regression analyses, other ethnicities were not considered even though these demographic groups are disproportionately represented in the night shift workforce[63]. Therefore, it is important to repeat these analyses in other ethnicities to reflect the multicultural population in the UK and other developed nations. Consistent with the current demographic of shift workers in the workforce, there were relatively fewer female than males undertaking shift work. This may explain why positive associations between day, mixed, night or permanent night shift work with SBP/DBP in males and unstratified datasets were absent in females.

Moreover, participants were generally older and had relatively high SBP/DBP (130 mmHg/80 mmHg); however, only a small percentage were diagnosed with hypertension or reported taking antihypertensive medication[64]. This is reflective of the demographics of the UK Biobank population, which is considered to be subject to "healthy volunteer" selection bias[65]. Characterisation of participants into subgroups based on average sleep length and quality, shift schedules and most other covariates were also extracted from self-reported data. We also acknowledge that some participants may experience symptoms but not clinically diagnosed with some health concerns such as sleep apnea or hypertension. Thus, we do not report on changes in risk of hypertension but rather positive

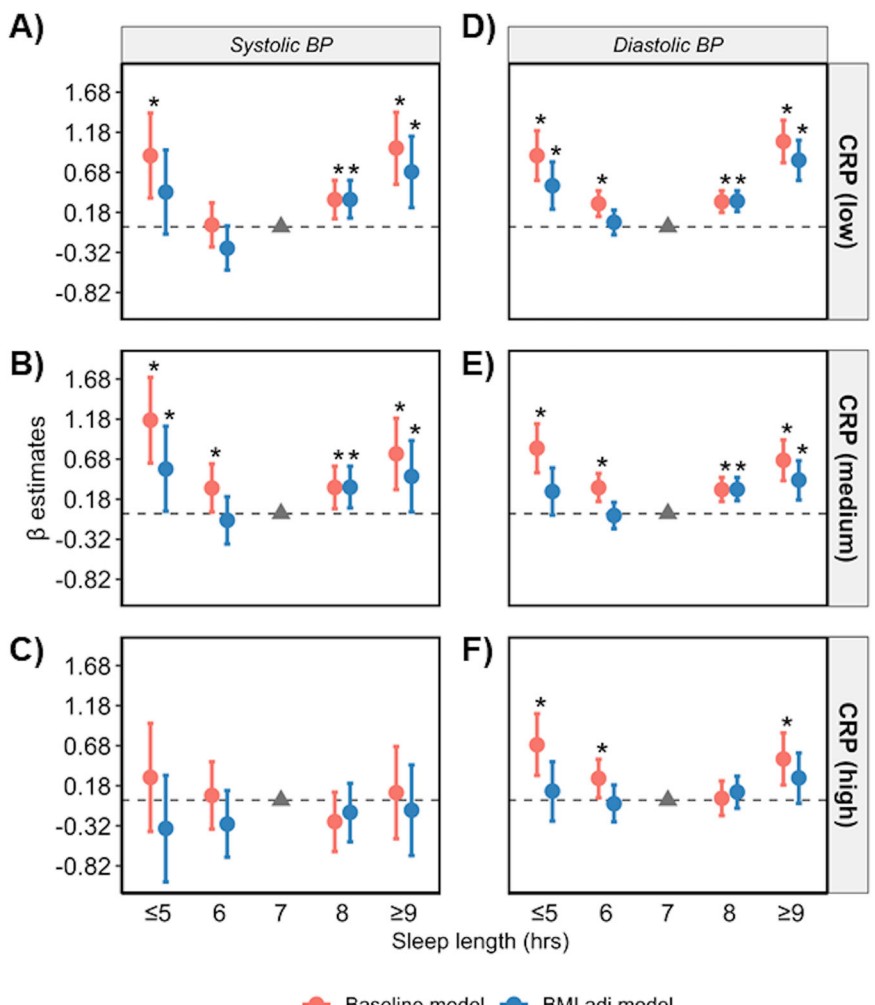

**Fig. 4 | Sleep length versus BP stratified by CRP levels.** Relationship between sleep length and SBP (**A**–**C**) and DBP (**D**–**F**) after data is stratified by low (<1 mg/L), medium (1–3 mg/L) and high (>3 mg/L) plasma CRP, respectively. Data are expressed as β-estimate ±95% CI, which are presented as the centre circle and corresponding error bars, respectively. *P*-values are estimated using multivariate logistic regression. $^*p < 0.05$ versus sleep length of 7 hr denoted by the solid grey triangle. Please see also Supplementary Table S13.

associations between sub-groups of circadian rhythm-disrupting behaviours and BP.

Despite these limitations, our findings demonstrate a persistent U-shaped relationship between sleep length and BP in the presence of low-grade inflammation. Importantly, the positive association between short and long sleep lengths and DBP were persistent in sub-groups of BMI, sex and age. These outcomes could have important implications for prevention strategies that can modify health risks within the population as well as for individuals. Previous studies have reported risk reductions of 21% and 34% for cardiovascular disease and stroke respectively, can be achieved in individuals by lowering BP by 5 mmHg[2,66,67]. Taken together our data suggest that implementation of measures that address lifestyle factors such as shift work, sleep quality and sleep lengths, could have similar cardioprotective benefits even with modest reductions in overall BP. The association between sleep length, BP and low-grade inflammation may reflect changes in factors that directly control BP, including vascular function, plasma renin, angiotensin II and aldosterone, or cytokine levels, which were not reported. Given that BP exhibits a strong 24 h profile that is clinically important, direct analysis of temporal rhythms of biomarkers of circadian disruption (i.e. cortisol or melatonin) and parallel assessments of sleep using actigraphy or polysomnography are required. Establishing changes in such parameters in individuals who experience acute and chronic episodes of circadian stress is essential to adequately understand the impact of disordered circadian rhythms on BP control over time.

In the present study, we demonstrated positive associations between short and long sleep lengths, compromised sleep quality or permanent night shift work with BP. We further showed that the U-shape relationship between sleep length and BP is strongest in non-shift workers and in a low-grade inflammatory status. Our findings highlight the adverse effects of circadian rhythm-disrupting behaviours on BP and a link with inflammation that is explored in a large and relatively unselected population. Maintaining appropriate sleep lengths and behaviours could be a non-conventional way to reduce risks of developing hypertension, particularly in shift workers.

## Methods

### Study population

This UK Biobank database (https://www.ukbiobank.ac.uk/) is a large population-based cohort study that recruited >500,000 individuals aged 40–70 years from across the UK (2006–2010). All participants provided written informed consent prior to any data collection. Ethical approval was obtained from the Northwest Multi-centre Research Ethics Committee (MREC) and the National Health Service (NHS) National Research Ethics Service on 17th June 2011 (Ref 11/NW/0382)

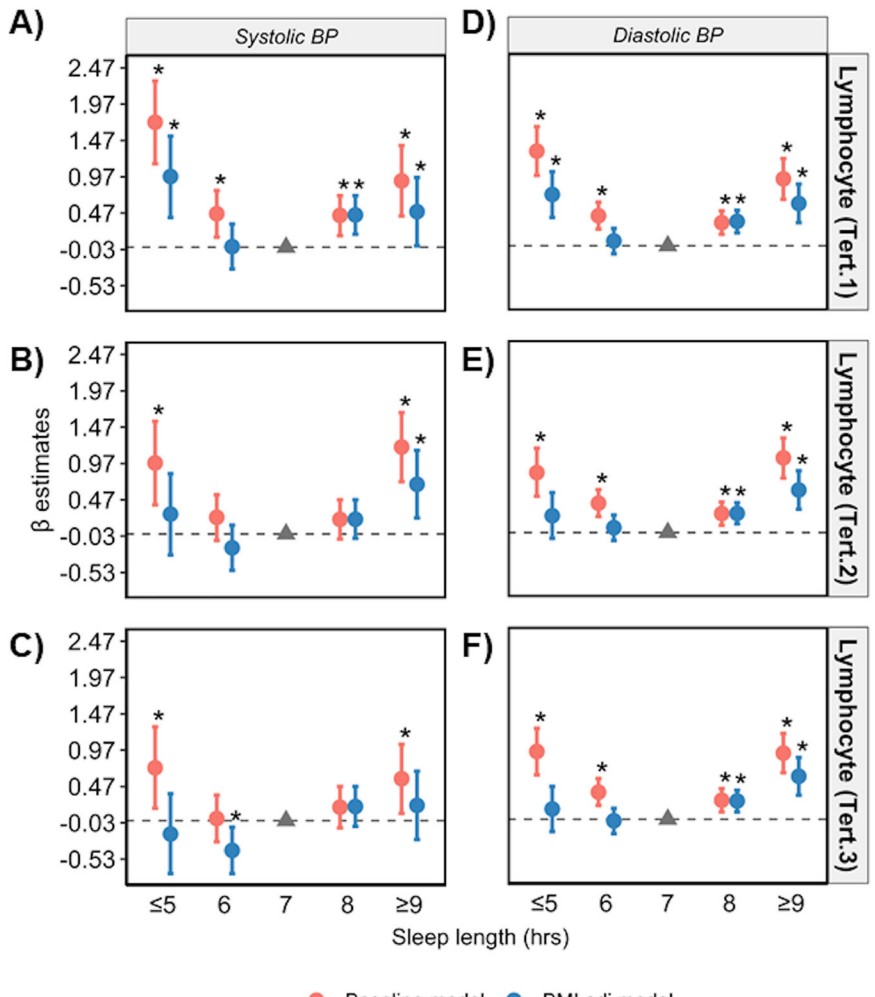

**Fig. 5 | Sleep length versus BP stratified by lymphocyte count.** Relationship between sleep length and SBP (**A**–**C**) and DBP (**D**–**F**) after data is stratified into lower (<1.62 × 10⁹ cells/L), middle (1.62–2.1 × 10⁹ cells/L) and upper (>2.1 × 10⁹ cells/L) terciles of lymphocyte counts, respectively. Data are expressed as β-estimate ± 95% CI, which are presented as the centre circle and corresponding error bars, respectively. *P*-values are estimated using multivariate logistic regression. * *p* < 0.05 versus sleep length of 7 hr denoted by the solid grey triangle. Please see also Supplementary Table S13.

and extended on the 10th May 2016 (Ref 16/NW/0274). This research has been conducted using the UK Biobank Resource under Application Number 55469. Analyses were restricted to participants with European white British ancestry (94% of the UK Biobank cohort) and provided familial and medical history, physiological measurements, and blood, urine and saliva samples. Baseline measurements and self-reported data collected at the time of recruitment were used. Participants using antidepressants, anti-psychotics, anxiolytics, and sleep medications were excluded (Supplementary Table S19)[68]. While many health and lifestyle factors can be considered circadian rhythm-disrupting behaviours, only 'sleep length', 'sleep quality' and 'shift work' were assessed for simplicity (Supplementary Fig. S12). Similarly, we acknowledge that gold standard markers of circadian disruption such as rhythms of body temperature, plasma cortisol, melatonin or BP have not been assessed to indicate the effects of circadian strain or misalignment on 24 h BP.

**Sleep length**

Participants were asked "About how many hours of sleep do you get in every 24 h (including naps)?" as an integer (hours per day). Participants were categorised into ≤5 h, 6, 7, 8 or ≥9 h groups based on their responses. Individuals diagnosed with sleep apnoea were removed from regression analyses relation to associations between sleep length and BP (Supplementary Table S20).

**Sleep quality score**

The sleep quality score was calculated using five sleep-related habits, including chronotype, sleep length, insomnia status, snoring and daytime sleepiness (Supplementary Table S20), to generate a cumulative value for each factor from 1 ("low risk") or 0 ("high risk"). Sleep quality was categorised as either "healthy" (4–5), "moderate" (2–3) or "poor" (0–1) (refer to Supplementary materials), which has been previously used in UK biobank cohort-based studies[69–72]. The healthy sleep quality category was the nominated reference category for all regression analyses related to sleep quality. Participants who selected "do not know" or "prefer not to answer" to any question were excluded for that sleep-related habit. Participants diagnosed with sleep apnoea were included in all statistical analyses related to sleep quality.

**Shift work**

Only participants who were employed at the time of recruitment were considered in shift work (Supplementary Table S20). Classification of shift work was drawn from responses to two questions, "Does your work involve shift work?" and "Does your work involve night shifts?" Shift work was defined in the UK Biobank as "work schedule that falls outside of the normal daytime working hours of 9AM–5PM", and night shift was specifically defined as "work schedule that involves working through the normal sleeping hours, for instance working through the

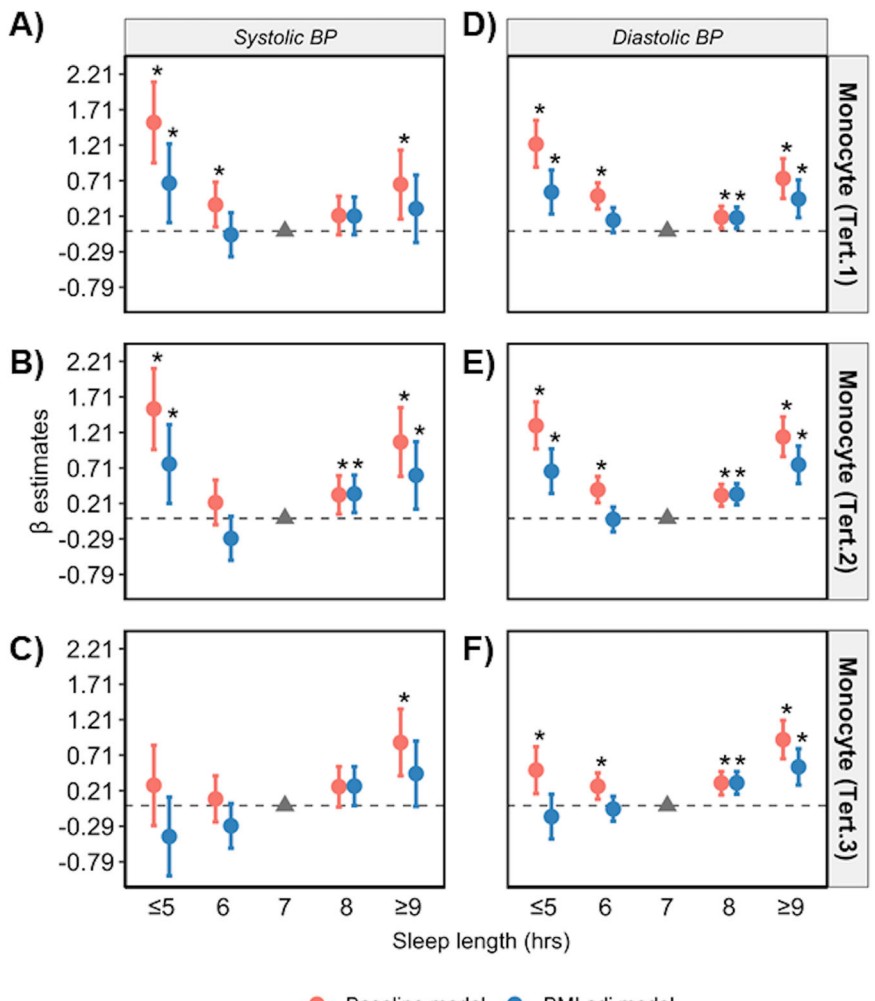

**Fig. 6 | Sleep length versus BP stratified by monocyte count.** Relationship between sleep length and SBP (**A**–**C**) and DBP (**D**–**F**) after data is stratified into lower (<0.4 × 109 cells/L), middle (0.4–0.51 × 109 cells/L) and upper (>0.51 × 109 cells/L) terciles of monocyte counts, respectively. Data are expressed as β-estimate ± 95% CI, which are presented as the centre circle and corresponding error bars, respectively. P-values are estimated using multivariate logistic regression. $^*p < 0.05$ versus sleep length of 7 hr denoted by the solid grey triangle. Please see also Supplementary Table S14.

hours from 12AM to 6AM." Participants were asked to select, "Never/Rarely", "Sometimes", "Usually" or "Always" for each question. The singular or combination of responses to both questions were used to allocate participants to either no shift work (reference category), day shift, mixed shift, night shift, or permanent night shift work (Supplementary Table S21).

## Assessment of BP
Mean SBP and DBP were calculated from two BP measurements. BP readings differing by >3 standard deviations away from the average of the differences between readings 1 and 2 were excluded. In participants without automated readings, mean SBP and DBP were determined from two manual BP measures where possible. ICD 10 codes, I10-I15, or ICD-9 codes, 401–405 were used to identify participants with a history of hypertension (Supplementary Table S20). Consumers of antihypertensive agents were identified if they reported taking any medications listed on the American Heart Association website (https://www.heart.org/en/; Supplementary Table S22). A systemic review and meta-analysis conducted by Paz et al. (2016) found that anti-hypertensive agents reduce SBP by 10–15 mm Hg and DBP by 8–10 mm Hg used provided as a monotherapy[73]. Therefore, we tested the upper and lower limits of these ranges as corrections for the use of

anti-hypertensives in this study. BP values for individuals reporting the use of anti-hypertensives were first adjusted by adding 15 and 10 mmHg to SBP and DBP, respectively, as performed in previous studies[17,74,75]. A second correction of adding 10 and 8 mmHg to SBP and DBP, respectively, found very similar results to the first correction. We, therefore elected to use the first correction (15 and 10 mmHg) as an adjustment in the baseline model.

## Systemic inflammation
A correlation plot was generated using neutrophil, monocyte, basophil, eosinophil and total leukocyte counts (10⁹ cells/L; Beckman Coulter LH750), and CRP (mg/L; Beckman Coulter AU5800) levels versus SBP, DBP, and sleep length, sleep quality and shift work categories (Supplementary Table S20).

## Covariates
Computed models considered sex, age, smoking status, alcohol intake, alcohol frequency, qualification (education), physical activity, history of depression and Townsend deprivation index as covariates (refer to Supplementary materials)[3,76,77]. Townsend deprivation index (UK Biobank) was used to classify socioeconomic status according to national census output for each postal code area.

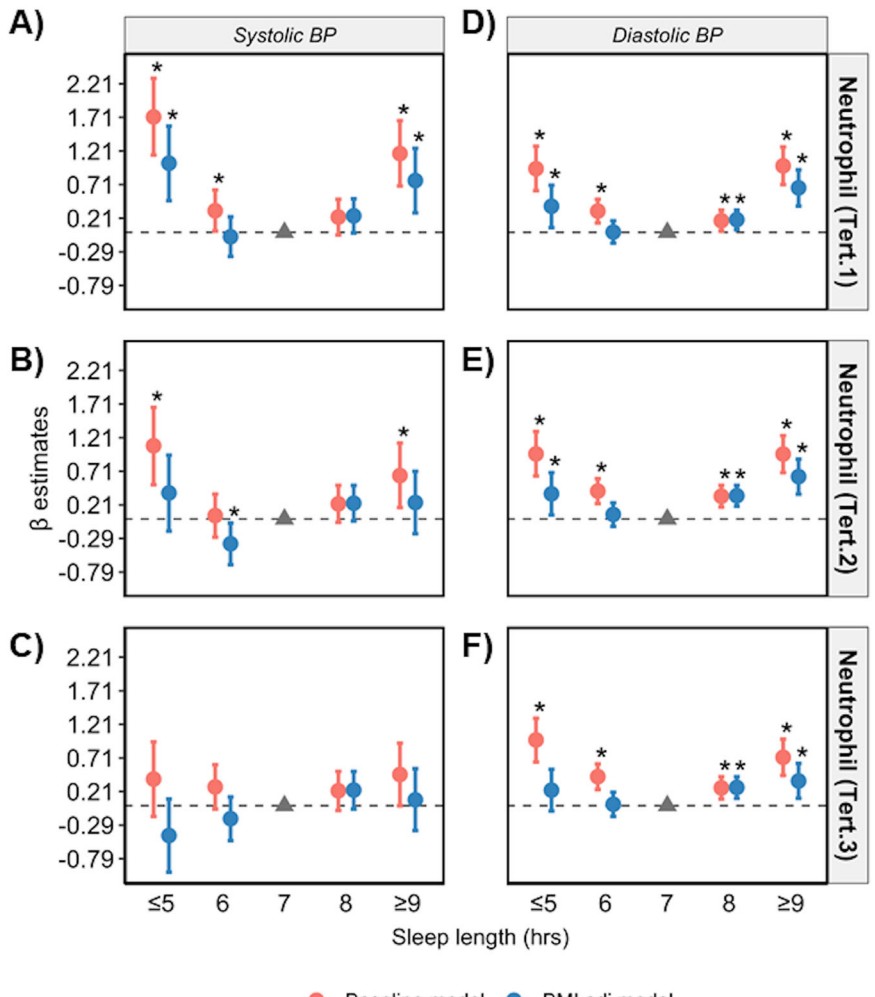

**Fig. 7 | Sleep length versus BP stratified by neutrophil count.** Relationship between sleep length and SBP (**A**–**C**) and DBP (**D**–**F**) after data is stratified into lower (<3.54 × 109 cells/L), middle (3.54–4.6 × 109 cells/L) and upper (>4.6 × 109 cells/L) terciles of neutrophil counts, respectively. Data are expressed as β-estimate ± 95% CI, which are presented as the centre circle and corresponding error bars, respectively. *P*-values are estimated using multivariate logistic regression. *p < 0.05 versus sleep length of 7 hr denoted by the solid grey triangle. Please see also Supplementary Table S14.

## GWAS summary-based MR

To test for potential causal relationships between sleep traits and BP, we used Single Nucleotide Polymorphisms (SNPs) as instruments for MR analyses. The motivation for conducting MR is to validate the association analyses conducted described above. To do this, we first obtained GWAS summary statistics for 8 sleep traits with a sample size of up to 700,000 people[26–28] and for BP with a sample size of up to 1 million people[29] (Supplementary Fig. S13). The GWAS of BP was also corrected for BMI; however, given that SNPs were used as instruments in MR, these analyses were expected to be independent of confounders. Of note, the GWAS for daytime sleepiness was the only sleep trait adjusted for BMI.

The sleep GWAS results included for daytime sleepiness[26] (answers to "How likely are you to dose off or fall asleep during the daytime when you don't mean to? (e.g. when working, reading or driving)" being coded as: 1 for "Never/rarely", 2 for "sometimes", 4 for "often" and 4 for all of the time, with or without adjustment of BMI, *n* = 452,071), sleep duration (hours, *n* = 446,118), long sleep duration (>8 h, *n* = 34,184 cases and 305,742 controls), short sleep duration (<7 h; *n* = 106,192 cases and 305,742 controls), frequent insomnia symptoms[27] (the answers to "Do you have trouble falling asleep at night or do you wake up in the middle of the night?" were "never/rare", "sometimes", or "usually" responding to the question; individuals

reported "usually" were considered as frequent insomnia symptom cases, *n* = 75,508 cases and 64,403 controls), morning chronotype[28] ("Definitely a 'morning' person", "More a 'morning' than 'evening' person", "More an 'evening' than a 'morning' person", "Definitely an 'evening' person", "Do not know" or "Prefer not to answer", which were coded as 2, 1, −1, −2, 0 and missing, respectively, *n* = 697,828) and day naps[26] (Day napping was coded continuously ("never/rarely", "sometimes", or "usually") responding to the question "Do you have a nap during the day?", *n* = 450,918). These consortia-based GWAS adjusted fixed effects including sex, age and 10 principal components from SNP genotypes[26–28]. The BP GWAS results included SBP and DBP, which are meta-analyses of the UK biobank (*n* = 458,557) and the International Consortium of BP Genome-Wide Association Studies (ICBP) (*n* = 299,024)[29]. After applying the GSMR to the above-mentioned summary data[30], we estimated the causal effect of sleep traits on BP traits. The 1000 Human Genomes were used as the reference to compute the SNP correlation matrix[78].

## Statistical analyses

Data processing, quality control and statistical analyses were performed using R Statistical Software (Version 3.6). The association between each sleep length, sleep quality, and shift work with SBP and DBP, was evaluated by multivariable linear regression analyses. For

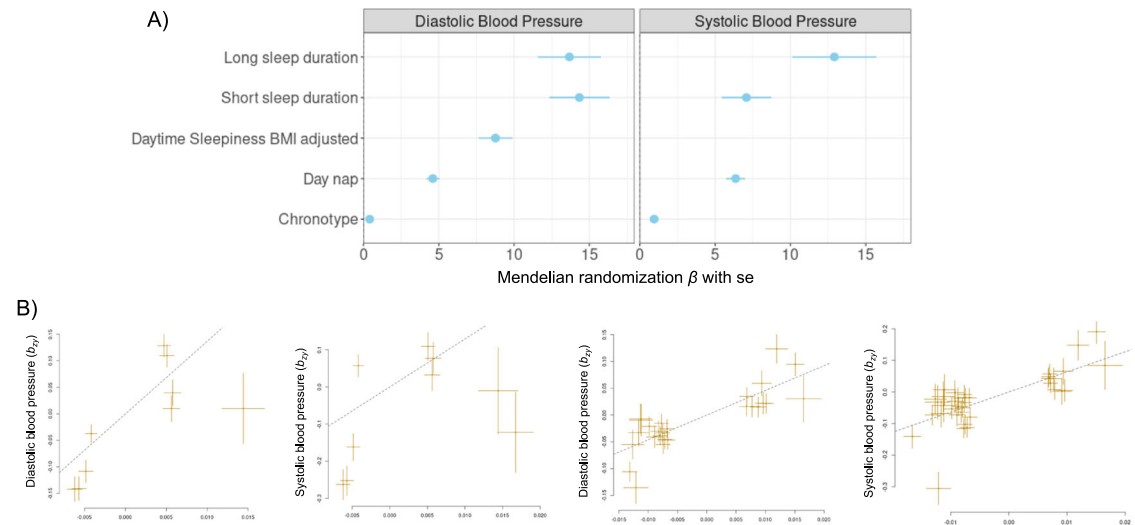

**Fig. 8 | Results of Mendelian Randomisation (MR) between sleep traits (exposure) and blood pressure (outcome) using GSMR. A** summary of all MR results. MR results are only shown if they had adjusted *p*-value < 0.05 and the results from GSMR agreed with MR-Egger and/or weighted median MR. Data are expressed as β-estimate ±95% CI, which are presented as the centre circle and corresponding error bars, respectively. **B** example of the effects (β-estimate) causal association between sleep traits and blood pressure. P-values of MR are based on the 2-sided test of GSMR after adjustment of multi-testing. Adjusted pMR = 6.4e-10 for long sleep duration and diastolic blood pressure. Adjusted pMR = 9.5e-6 for long sleep duration and systolic blood pressure. Adjusted pMR = 8.7e-25 for day nap and diastolic blood pressure. Adjusted pMR = 4e-23 for day nap and systolic blood pressure. Please see also Table 2.

sleep length and sleep quality, the following models were fitted: (i) 'baseline model', adjusting for sex, age, smoking status, alcohol intake and frequency, qualification, employment status, depression, physical activity and Townsend deprivation index; (ii) 'BMI adj model', baseline model adjusted for BMI; (iii) 'Antihypertensive (AH) free model', excluding participants reporting the use of antihypertensives in the baseline model; (iv) 'AH free BMI adj model', AH free model further adjusted for BMI; (v) 'CRP adj model', baseline model with adjustment for CRP; (vi) 'BMI and CRP adj model', baseline model further adjusting for both CRP and BMI. Analyses for shift work and BP were fitted to identical models; however, were restricted to include employed individuals only and not adjusted for a history of depression.

Each model was fitted to subgroups stratified by BMI (< 25, 25-< 30 and ≥30 kg/m²), sex (male and female), age (≤50 and >50 years) and low, medium and high levels of CRP (<1, 1–3 and >3 mg/L) and terciles of lymphocyte (<1.62, 1.62–2.1 and >2.1 $10^9$ cells/L), monocyte (<0.4, 0.4–0.51 and >0.51 $10^9$ cells/L), and neutrophil (<3.54, 3.54–4.6 and >4.6 $10^9$ cells/L) counts. Data are expressed as the beta (β)-estimate value ± 95% confidence interval (95% CI) for changes in SBP or DBP compared to the reference category. $p < 0.05$ was considered statistically significant.

To identify independent SNP instruments for each exposure (sleep traits), GWAS-significant SNPs ($p_{GWAS} < 5 \times 10^{-8}$) were pruned ($r^2 < 0.01$; linkage disequilibrium (LD), within windows of 10,000 kb). For the MR analysis between long sleep duration and BP, there were no SNPs left after filtering for $r^2 < 0.01$. Therefore, the MR analysis between long sleep duration and BP used 9 SNPs pruned at $r^2 < 0.1$, which has been accepted in conducting MR between sleep traits and cardiovascular diseases[24,25]. These SNP selections had F-statistics ranging from 34.2 to 51.2 (Table 2), which are well above the accepted threshold of 10[79]. The HEIDI test implemented in GSMR was used to detect and remove ($p_{HEIDI}<0.01$) variants with horizontal pleiotropy (i.e., having independent effects on both sleep traits and BP), as they were not underlying assumptions for valid instruments. $p_{GSMR}$ was adjusted by FDR. We also conducted additional sensitivity analyses using MR-Egger and weighted median approaches implemented in the R package MendelianRandomization[31]. MR results were considered significant if they had FDR-adjusted $p_{GSMR} < 0.05$ in analyses of MR-

Egger or weighted median and had a consistent effect direction across the three methods. SNPs used in the additional MR analysis were identical to those used for GSMR as described above. Three established MR methods are used to ensure that 1) tested SNPs are associated with the exposure ($p < 5e-8$), 2) SNPs are not associated with confounder (e.g., BMI were adjusted), and 3) SNPs are expected to influence the blood pressure via sleep disruptions.

We also performed non-linear MR between sleep duration and BP using two methods. The assumption of the potential existence of non-linear causal relationships was based on the above-described results: 1) significant association analyses between sleep duration and blood pressure and 2) significant linear MR analyses between sleep phenotypes and blood pressure. We first used PolyMR[80], which has a flexible non-linear relationship assumptions and requires raw data and conveniently fits multi-SNP genotype data as instrumental variables. Therefore, SNP genotypes and raw phenotypes from UK Biobank used in the linear regression analyses described above were used. The genotype data of 33 SNPs that were significant in GWAS ($p < 5e-8$) for both sleep duration and BP (presented in the UK Biobank) were extracted and read into PolyMR as instrumental variables. Before MR analysis, BP was adjusted for covariates used in the linear regression analysis and the residual of the outcome variable was fit into PolyMR using default options. The built-in plotting function (plot_polymr) was used to visualise the results. Secondly, we performed non-linear MR (doubly-ranked stratification method)[81] using the software implemented in SUMnlmr[33]. The model assumption of doubly-ranked stratification method[81], which does not require strict parametric assumptions to create strata of the population that have different average levels of exposure is suitable as the exposure of sleep duration is an ordinal variable. As SUMnlmr only accepts one slot for instrumental variables, we used those 33 SNPs to generate polygenic risk score (PRS) using the plink-score[82] function for MR analysis in SUMnlmr. As outlined in Burgess et al.[83], we fitted covariates used in the linear regression analyses into SUMnlmr and performed non-linear MR using default options. Results were plotted following instructions on https://github.com/amymariemason/SUMnlmr. The estimates of localised average causal effects (LACE) from non-linear Mendelian Randomisation using SUMnlmr as Supplementary Table S23.

**Table 2 | Mendelian randomisation (MR) β-estimates using GSMR for sleep length and sleep quality trains and BP**

| BP | Sleep trait | β-estimate | SE | p-value | SNP count | FDR-adjusted p-value of MR | F-statistic | β-estimate (MR-Egger) | SE (MR-Egger) | p-value (MR-Egger) | β-estimate (Weighted median) | SE (Weighted median) | p-value (Weighted median) |
|---|---|---|---|---|---|---|---|---|---|---|---|---|---|
| Systolic BP | Sleep duration | -0.106 | 0.378 | 0.779 | 34 | 0.779 | 37.31 | 2.32 | 2.320 | 0.203 | 0.611 | 0.611 | 0.467 |
| | Short sleep duration | 7.078 | 1.645 | 1.70E-05 | 10 | 2.37E-05 | 38.22 | 10.437 | 10.437 | 0.011 | 2.460 | 2.460 | 0.163 |
| | Long sleep duration | 12.912 | 2.785 | 3.54E-06 | 9 | 9.45E-06 | 35.88 | 22.55 | 22.550 | 0.981 | 5.334 | 5.334 | 0.007 |
| | Insomnia | -3.882 | 0.806 | 1.45E-06 | 15 | 2.26E-06 | 50.95 | 2.283 | 2.283 | 0.081 | 1.135 | 1.135 | 1.00E-04 |
| | Daytime sleepiness (adjusted for BMI) | -2.252 | 1.606 | 0.161 | 8 | 0.173 | 40.21 | 9.592 | 9.592 | 0.728 | 2.275 | 2.275 | 0.328 |
| | Daytime sleepiness | 15.788 | 1.984 | 1.74E-15 | 6 | 8.11E-15 | 42.53 | 5.925 | 5.925 | 0.645 | 1.419 | 1.419 | 0.138 |
| | Daytime napping | 6.359 | 0.630 | 5.71E-24 | 37 | 4.00E-23 | 43.66 | 3.282 | 3.282 | 0.009 | 0.979 | 0.979 | 1.66E-08 |
| | Chronotype | 0.949 | 0.132 | 7.46E-13 | 57 | 1.59E-12 | 44.4 | 0.572 | 0.572 | 0.043 | 0.230 | 0.230 | 0.006 |
| Diastolic BP | Sleep duration | 0.509 | 0.217 | 0.019 | 34 | 0.024 | 37.18 | 1.517 | 1.517 | 0.555 | 0.357 | 0.357 | 0.694 |
| | Short sleep duration | 14.337 | 2.002 | 7.95E-13 | 3 | 1.59E-12 | 48.25 | 9.833 | 9.833 | 0.036 | 2.295 | 2.295 | 8.65E-12 |
| | Long sleep duration | 13.672 | 2.103 | 7.96E-11 | 9 | 6.37E-10 | 34.15 | 14.653 | 14.653 | 0.987 | 2.848 | 2.848 | 2.26E-14 |
| | Insomnia | -0.665 | 0.471 | 0.158 | 14 | 0.173 | 51.23 | 1.433 | 1.433 | 0.897 | 0.685 | 0.685 | 0.549 |
| | Daytime sleepiness (adjusted for BMI) | 8.766 | 1.126 | 7.08E-15 | 6 | 2.48E-14 | 40.19 | 14.051 | 14.051 | 0.339 | 1.843 | 1.843 | 2.79E-05 |
| | Daytime sleepiness | 6.829 | 0.901 | 3.43E-14 | 9 | 9.61E-14 | 43.53 | 3.141 | 3.141 | 0.999 | 0.804 | 0.804 | 0.135 |
| | Daytime napping | 4.607 | 0.437 | 6.18E-26 | 26 | 8.66E-25 | 43.69 | 2.201 | 2.201 | 0.002 | 0.687 | 0.687 | 1.70E-10 |
| | Chronotype | 0.412 | 0.078 | 1.29E-07 | 55 | 2.26E-07 | 43.63 | 0.313 | 0.313 | 0.409 | 0.134 | 0.134 | 0.004 |

## Data availability

This research was conducted using the UK Biobank Resource (https://www.ukbiobank.ac.uk/) under application number 55469. Data from UK Biobank is accessible to eligible researchers via applying to www.ukbiobank.ac.uk. Data supporting the findings of this study are available in the article and its Supplementary information. Source data and a table of all n values for each subgroup analysis are provided as Source Data files and have also been deposited in Figshare accession code https://doi.org/10.6084/m9.figshare.24062760[84]. Beta estimates for all data presented as figures in the main paper and the supplementary section are presented in tables in the supplementary data section. The sleep GWAS summary data used for Mendelian Randomisation are publicly available and were obtained from the Sleep Disorder Knowledge Portal (http://sleepdisordergenetics.org/, PMID:30696823, PMID:30804566 and PMID:31409809)[26–28]. The BP GWAS summary data were obtained from International Consortium of BP Genome-Wide Association Studies (ICBP)[29] via GWAS Catalogue (accessions: GCST90132904 and GCST90132903). Source data are provided with this paper.

## Code availability

The analyses used public software R (Version 3.6) for linear regression, GSMR (https://yanglab.westlake.edu.cn/software/gsmr/)[30] and R package of MendelianRandomization (https://cran.r-project.org/web/packages/MendelianRandomization/index.html)[31] for linear Mendelian Randomisation; PolyMR (https://github.com/JonSulc/PolyMR)[32] and SUMnlmr (https://github.com/amymariemason/SUMnlmr)[33] for non-linear Mendelian Randomisation. All other codes for the analyses are available at https://github.com/rxiangr/UKB_Sleep_and_Blood_Pressure/tree/main.

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

## Acknowledgements

This research was conducted using the UK Biobank Resource (https://www.ukbiobank.ac.uk/) under application number 55469. This study was supported in part by the Research Training Program (RTP) Stipend PhD scholarship from Monash University (2020) (MK) and M.J.Y. is supported by the Alice Baker and Eleanor Shaw Gender Equity Fellowship, Baker Trustees. The Baker Heart and Diabetes Institute and Hudson Institute of Medical Research are supported by the Victorian Government's Operational Infrastructure Scheme.

## Author contributions

M.K. performed the study, interpreted the data, drafted the manuscript. A.N. performed the study, collected and analysed the data, drafted the manuscript. R.X. performed the MR, reviewed and contributed to preparation of the manuscript. S.Y. interpreted the data, reviewed the manuscript. P.F. reviewed the manuscript. T.C. reviewed the manuscript. R.C. analysed and interpreted the data, reviewed the manuscript. M.R. conceived the project, interpreted the data, drafted the manuscript.

## Competing interests

The authors declare no competing interests.
