## [Peer Review File · Nature Communications]

REVIEWER COMMENTS

Reviewer #1 (Remarks to the Author):

This is a study combining both observational and Mendelian randomization design in assessing different sleep traits with hypertension in UK Biobank and GWAS summary statistics. Whilst understanding sleep's effect is of global health importance, this has been explored in previous studies using the same data. There is also room for improvement for the MR part, where addition of non-linear MR would improve overall novelty. Please see below for my comments.

Major comments

Please provide more justification regarding proposed effect modification by sex, age, BMI, and inflammation.

There has been previous studies using the same cohort to address related questions. Please clarify how your study can add to previous literature using the same data apart from sample size difference (PMID: 34869684; 35876003), as well as other Mendelian randomisation studies (PMID: 33822910; 34315237)

There should be more description on the GWAS used, such as covariable adjustments, and the units of the phenotypes. For morning chronotype, what does it mean by 4-level continuous intervals? How does this impact the interpretation of the gene-phenotype estimates?

Please consider using an r^2 of 0.001 to ensure the instruments are independent.

Whilst including an MR component to verify causal relation is good, the current MR analysis is likely too brief. Including additional sensitivity analyses such as MR-Egger and weighted median would be necessary, as well as reporting strength of instruments (F statistics). Including non-linear MR approach using the doubly ranked stratification method would greatly improve novelty of findings (<https://www.biorxiv.org/content/10.1101/2022.06.28.497930v1>) as some studies suggest the residual method used in earlier non-linear MR studies (not on sleep as this was not assessed) could be biased (PMID: 36528346)

One main limitation is possible confounding in observational analyses. I think this is a cross sectional analyses and hence reverse causation could be an issue. Although MR analyses suggested possible

associations, the lack of non-linear MR is a main issue, especially when the observational analysis suggested potential non-linearity.

Minor comments

Line 151: Please explain the purpose of drawing the correlation plot.

Line 158: Township index? Or do you mean Townsend deprivation index?

Please move the statistical analyses under Mendelian randomisation to statistical analyses

Please insert back the reference group in the tables (e.g. supplemental table 5)

Line 412: What do the authors mean by "have contributed to confounding effects of adjusting for BMI".

Reviewer #2 (Remarks to the Author):

Peer review report

Thank you for the opportunity to review this interesting paper. The aim of the authors was to assess the associations between sleep length, sleep quality, shift work, inflammatory markers and BP and determine whether these associations differed across BMI, age, sex categories. The topic is relevant and important given the health and socioeconomic burden of hypertension and sleep problems. Overall, the aims of the study were achieved, and the paper is well written and structured.

Key results

The results are well presented and show that sleep length, sleep quality, and shift work are significantly associated with changes in BP.

Validity

The use of a nationally representative dataset and large sample size makes the findings from this study more robust. However, my suggestion is that the authors make it clear from the beginning of the

manuscript that the data was restricted to adults of European white British ancestry; this should appear in the abstract section at the very least, if not the title

Significance

This study findings support other prior findings showing that problems with sleep, including poor sleep quality, extremes in sleep duration, and working schedules that affect circadian rhythms are associated with adverse health outcomes including increased risk for hypertension. This is especially important given the increasing evidence suggesting that sleep health disparities may be major contributors to other health disparities

Data, Methodology and Analytical approach

The use of multivariable linear regression models to test associations between the sleep parameters, shift work, BP measures were appropriate based on the research objectives of the study. However, I am unable to evaluate the use of GWAS summary based Mendelian randomization (MR) to test for causation since this is beyond the scope of my expertise.

Just a few comments:

1. Complete the data availability statement “summary data were obtained from....”
2. Sleep quality score: was the scoring based on a validated scale/tool or this is a new way of scoring sleep quality developed by the authors? I request the authors to provide reference if the score has been used previously.
3. Statistical analyses: AH free model: I have an idea of what AH stands for, but the authors should make that clear for the readers
4. It would be helpful if the authors would outline their justifications for stratification based on age, inflammatory markers, sex, BMI among others. There is a statement in introduction section “previous studies have not evaluated associations stratified for BMI, sex, age or inflammation” but more should be provided.

Results section

The results are well presented and reflect the objectives of the study. I have a few comments below which the authors can consider:

- The authors should make it clear in the main text which model they are presenting results on. For example, from Lines 224 – 230 the model is not stated
- Since a lot of results presented in the main text refer to supplementary tables and figures, I would rather the authors use statistical terms (significant/ not significant) or/and include p-values, rather than

using terms such as “strongest”, “modest” when presenting results. This would make it clear to the reader what associations are statistically significant and which are not.

- Would the authors clarify this statement in Line 301 “which reflects the strong correlation between BMI and CRP”
- Table 2: Improve the readability of this table especially the way p-value are presented.

Suggested improvements

No additional analyzes are suggested. For other suggestions/comments, see above

Clarity and context

The findings are clearly presented, and the findings have been discussed well in the context of previous research

References

No obvious problem noted with references, The manuscript references previous literature appropriately.

Your expertise

I am unable to sufficiently evaluate the use of GWAS summary based Mendelian randomization (MR) to test for causation since this is beyond the scope of my expertise.

Reviewer #3 (Remarks to the Author):

Kanki and colleagues demonstrated that sleep duration has a U-shaped association with BP, such that generally, anything on either side of 7h/24 h increases BP. Similar associations exist between sleep quality and night shift work, with permanent shift work and the poorest sleep quality showing a risk for increased BP. Furthermore, there was a U-shape relationship between sleep duration and BP with strongest relationship in people with low CRP. This manuscript must is a lot of hard meticulous work and the authors should be congratulated for these analyses.

Some issues are noted here:

Abstract

The authors state that they examine the association between markers of circadian dysregulation, BP and inflammatory markers. There is no examination of circadian dysregulation except for in shift-workers. Therefore, this term should be removed. The authors conclude that they have demonstrated an “independent association between circadian rhythm-disrupting behaviours, adverse SBP/DBP regulation and increased inflammatory markers in males and females.” If the association between sleep duration and BP was the strongest in people with CRP levels <1 mg/L, this would indicate that the association is strongest in people with NO inflammation and not increased inflammatory markers.

Introduction:

The authors introduce the importance of diurnal BP patterns and cardiovascular health. There is no examination of diurnal patterns of BP. The measurements for BP were in duplicate, but it is not known when they were made or the details of how they were obtained. Therefore, it may be prudent to focus the introduction on just office BP. As stated above, in this reviewer’s opinion, other than shift work, there is no evidence for circadian strain because the questions about sleep were over 24h. For instance, the circadian ‘disruption’ is different if a person sleeps from, for e.g., 9 PM to 6 AM vs someone who sleeps 10:30 PM-6 AM with 1.5h nap in the afternoon when melatonin concentrations are negligible. Therefore, circadian strain cannot be determined. Therefore, in my opinion, the focus should be on sleep and shift work, rather than circadian disruption or strain.

Methods:

Sleep apnea is quite pervasive and people may not know they have it. Simply excluding people with a h/o sleep apnea would not exclude people with sleep apnea.

It would be good to state why the 7 h group was considered as standard/ control and not the 8 h group, or both groups combined together.

Adding 15 and 10 mmHg to people taking BP meds seems too arbitrary. The authors cite one paper who did that, who in turn had cited another paper which state that where this method may be used in clinical trials. That paper also stated that there is virtue in using 10 mm for SBP. The authors should do a sensitivity analysis with both these numbers. Additionally, the removal of participants with HTN meds produced similar, but not identical results with sleep quality. Plus, this was not necessarily a clinical trial.

Results:

This title: "Sleep length and sleep quality are associated with a significant change in BP" should perhaps be rephrased as there is no examination of a 'change' in BP.

Because the sample was predominantly white, the conclusions should say so as these may not be generalizable to other groups (as the authors correctly point out).

Discussion:

In this reviewer's opinion, any statement about circadian strain should be edited to focus on sleep and shift work. The authors discuss elevation of BP, but there is no change studied; therefore, these statements should be toned down. The authors should discuss in detail about people with normal CRP levels, i.e., presumably no clinical inflammation, have the strongest relationship between sleep duration and BP. It would be educational to the readers to see a scenario translating the beta values to an actual number. For instance, if someone's BP increased by 2 mm because of sleeping for longer or shorter time, even if it may be statistically significant, it may not be clinically meaningful, especially in people who are already taking HTN meds. Taken together the data support an independent link between sleep duration, and BP, but not circadian strain and increased risk of hypertension, especially since this was not a longitudinal study examining people who were normotensive and later became hypertensive.

REVIEWER COMMENTS

Reviewer #1 (Remarks to the Author):

This is a study combining both observational and Mendelian randomization design in assessing different sleep traits with hypertension in UK Biobank and GWAS summary statistics. Whilst understanding sleep's effect is of global health importance, this has been explored in previous studies using the same data. There is also room for improvement for the MR part, where addition of non-linear MR would improve overall novelty.

We thank the Reviewer for their feedback and agree with their suggestions to improve upon our current Mendelian randomization (MR) analyses, which we have undertaken in this revision. We also believe performing the proposed linear and non-linear MR analyses have significantly enhanced the quality and novelty of the data in this manuscript compared to recent publications using the UK Biobank database. Please see our responses to the major and minor comments outlined below.

Major comments

Please provide more justification regarding proposed effect modification by sex, age, BMI, and inflammation.

To the best of our knowledge, the effects of covariates such as sex, age, BMI and inflammation in the current literature are handled by adjusting for such factors in regression analyses. However, there is evidence that the impact of circadian disruption can be dependent on sex, age, BMI and inflammatory status. Furthermore, inflammatory responses are strongly gated by the circadian clock and glucocorticoid levels, both of which can become dysregulated in settings of circadian misalignment. The output variable, blood pressure, also differs between males and female, is age-dependent, and is strongly associated with obesity. Recent studies have also demonstrated a link between elevated circulating inflammatory factors including C-reactive protein levels in plasma, white blood cell counts and circulating cytokine factors and an increased blood pressure. Therefore, we have tested the effect of both adjusting for these covariates as well as detailed subgroup analyses for each covariate. Furthermore, we therefore sort to determine if the positive association between circadian rhythm-disrupting behaviours and blood pressure could be linked to systemic markers of inflammation.

To highlight these points and justify stratifying by or adjusting for key covariates, sex, age, BMI and inflammation, we have edited the Introduction and Discussion sections accordingly.

We have summarised these points in the Introduction:

“Chronic low-grade inflammation is also an important contributor to the pathophysiology of hypertension. Biomarkers of systemic inflammation, such as plasma C-reactive protein (CRP) levels and white blood cell counts, are associated with high BP and mortality related to cardiovascular diseases. Furthermore, inflammatory responses are strongly gated by the circadian clock and glucocorticoid levels, both of which can become dysregulated in settings of circadian misalignment. While shift work, reduced sleep and an evening chronotype, are independently associated with higher levels of circulating inflammatory markers, whether inflammatory factors are linked to associations between circadian rhythm-disrupting behaviours and BP has not been evaluated. Similarly, previous studies have not evaluated such associations when data are stratified for BMI, sex, age or inflammatory status, and even fewer have evaluated the impact of shift work on blood pressure in a cohort of this size.”

Page 3 lines 74 – 84

We have also extended the Discussion section to include:

“Sex, age, BMI and inflammation are some of the key covariates that have been linked to the adverse impact of circadian dysregulation on cardiovascular disease.²² For example, the prevalence of cardiovascular disease and risk factors including hypertension dramatically increases in post-menopausal women to match that of men, and survivorship following time of day-dependent myocardial infarction also shows sexual dimorphism.^{23, 24} Furthermore, the robustness of biological circadian rhythms exhibits age-dependent declines and can be blunted with higher BMI and pro-inflammatory markers.^{25, 26, 27} The size of the UK Biobank dataset enabled detailed subgroup analyses, which further revealed that this positive association between non-standard sleep length (< 7hr <) and BP was largely independent of established demographics including sex, age and BMI.

Page 10 – 11 lines 277 - 286

There have been previous studies using the same cohort to address related questions. Please clarify how your study can add to previous literature using the same data apart

from sample size difference (PMID: 34869684; 35876003), as well as other Mendelian randomisation studies (PMID: 33822910; 34315237).

We thank the Reviewer for raising the importance of clearly establishing the novelty of our manuscript in comparison to other studies that have also used the UK Biobank database and MR approaches. To address this, we have highlighted key differences between our study and those mentioned above.

- 1) PMID 35876003 – Yang et al. (2022) Hypertension. “Association of Nap Frequency with Hypertension or Ischemic Stroke Supported by Prospective Cohort Data and Mendelian Randomization in Predominantly Middle-Aged European Subjects”***

The above study only investigates the associations between selected sleep disturbances (daytime napping and nap frequency) with the risk of hypertension or ischemic stroke. In our current study, we evaluated the effect of multiple sleep-related habits including insomnia, snoring, and chronotype as well as sleep duration to give a clearer overall assessment of sleep quality. Furthermore, the above study, like many others, only considers adjusting for common covariates (e.g., sex, age, BMI etc.) but do not assess key confounding effects using stratified datasets, e.g., blood cell counts, as we have done in our manuscript.

- 2) PMID 34869684 – Li et al. (2021) Frontiers in Cardiovascular Medicine. “Healthy Sleep Associated with Lower Risk of Hypertension Regardless of Genetic Risk: A Population-Based Cohort Study”***

The linear regression models performed in this study only adjust for the covariates of age, sex, BMI and other common confounders such smoking. Although, a very similar variation of the sleep quality score is used, a more sensitive breakdown of sleep lengths, plus the inclusion of shift work, were assessed in our study. In addition, we also stratified the analyses into different blood cell counts which are not done in the cited literature.

- 3) PMID 33822910 – Ai et al. (2021) European Heart Journal. “Causal associations of short and long sleep durations with 12 cardiovascular diseases: linear and nonlinear Mendelian randomization analyses in UK Biobank”***

While the datasets in the above study present causal inferences using linear and non-linear MR analyses, the outcomes differ between this study and ours. The authors evaluated the genetically predicted sleep length (PRS) and the incidence of 12 cardiovascular diseases, including the incidence of hypertension. Note that the PRS used by Ai et al. (2021) explained <1% of the variance in sleep duration. In contrast, using raw and/or summary data of multiple sleep traits, we identified that short/long sleep lengths, poor & moderate sleep qualities and permanent night-shift work are each associated with higher blood pressure compared to the respective reference categories. Our MR analysis also detected significant linear relationships and suggestive non-linear relationships between sleep traits and blood pressure.

- 4) PMID 34315237 – Liu et al. (2021) Journal of the American Heart Association. “Genetically Predicted Insomnia in Relation to 14 Cardiovascular Conditions and 17 Cardiometabolic Risk Factors: A Mendelian Randomization Study”*

This study investigates the causal effects of genetically predicted-insomnia and the risk of mediators of cardiovascular diseases. In contrast, we investigated the causal associations between insomnia, chronotype, sleep length, day time sleepiness and blood pressure using linear MR analyses. In the above study insomnia was positively associated with only DBP, and this was absent when analyses were further adjusted for BMI.

We further emphasise that the causal relationship between sleep duration/quality and blood pressure has been largely obtained from using gold standard PSG and measures of melatonin in rigorously controlled laboratory-based studies. These acute studies are limited in sample size and preclude analysis of causality at the population level. Similarly, stratifying these datasets according to key confounding variables for analysis of associations is limited. The UK Biobank is one of the largest cohorts of its kind and has allowed us to evaluate linear regression models within stratified datasets and both linear and non-linear MR analyses. This has enabled us to comprehensively evaluate potential moderators in a large sample size, and identify (to an extent) causality inferences on a population-based scale.

To summarise the points above, we have edited our Introduction and Discussion, respectively, to include:

“While several recent studies provide support for a relationships between sleep phenotypes and cardiovascular diseases, these studies did not undertake detailed stratification of the data, or focused on one sleep trait only or investigated genetically

predicted sleep duration. The novelty of the current study was therefore to determine whether the positive association between circadian rhythm-disrupting behaviours including multiple sleep phenotypes and BP could be linked to systemic markers of inflammation.”

Page 4 lines 90 - 96

“Moreover, participants were generally older and had relatively high SBP/DBP (130mmHg/80mmHg); however, only a small percentage were diagnosed with hypertension or reported taking antihypertensive medication. This is reflective of the demographics of the UK Biobank population, which is considered to be subject to “healthy volunteer” selection bias. Characterisation of participants into subgroups based on average sleep length and quality, shift schedules and most other covariates were also extracted from self-reported data. We also acknowledge that some participants may experience symptoms but not clinically diagnosed with some health concerns such as sleep apnea or hypertension. Thus, we do not report on changes in risk of hypertension but rather positive associations between sub-groups of circadian rhythm-disrupting behaviours and BP.”

Page 13 – 14 Lines 391 - 400

There should be more description on the GWAS used, such as covariable adjustments, and the units of the phenotypes. For morning chronotype, what does it mean by 4-level continuous intervals? How does this impact the interpretation of the gene-phenotype estimates?

We have updated the Methods section related to GWAS data as following:

“To test for potential causal relationships between sleep traits and BP, we used Single Nucleotide Polymorphisms (SNPs) as instruments for MR analyses. To do this, we first obtained GWAS summary statistics for 8 sleep traits with a sample size of up to 700,000 people and for BP with a sample size of up to 1 million people (Supplementary Figure S13). The GWAS of BP was also corrected for BMI; however, given that SNPs were used as instruments in MR, these analyses were expected to be independent of confounders. Of note, the GWAS for daytime sleepiness was the only sleep trait adjusted for BMI.

The sleep GWAS results included for daytime sleepiness (answers to “How likely are you to dose off or fall asleep during the daytime when you don’t mean to? (e.g. when working, reading or driving)” being coded as: 1 for “Never/rarely” , 2 for “sometimes”,

4 for “often” and 4 for all of the time, with or without adjustment of BMI, n = 452,071), sleep duration (hours, n = 446,118), long sleep duration (>8 hours, n = 34,184 cases and 305,742 controls), short sleep duration (<7 hr; n = 106,192 cases and 305,742 controls), frequent insomnia symptoms. (the answers to “Do you have trouble falling asleep at night or do you wake up in the middle of the night?” were “never/rare”, “sometimes”, or “usually” responding to the question; individuals reported “usually” were considered as frequent insomnia symptom cases, n = 75,508 cases and 64,403 controls), morning chronotype (Definitely a ‘morning’ person”, “More a ‘morning’ than ‘evening’ person”, “More an ‘evening’ than a ‘morning’ person”, “Definitely an ‘evening’ person”, “Do not know” or “Prefer not to answer”, which were coded as 2, 1, -1, -2, 0 and missing, respectively, n = 697,828) and day naps (Day napping was coded continuously (“never/rarely”, “sometimes”, or “usually”) responding to the question “Do you have a nap during the day?”, n = 450,918). These consortia-based GWAS adjusted fixed effects including sex, age and 10 principal components from SNP genotypes. The BP GWAS results included SBP and DBP, which are meta-analyses of the UK biobank (n=458,557) and the International Consortium of BP Genome-Wide Association Studies (ICBP) (n=299,024). After applying the GSMR to the above-mentioned summary data, we estimated the causal effect of sleep traits on BP traits. The 1000 Human Genomes were used as the reference to compute the SNP correlation matrix.”

Page 18 – 19 lines 509 - 537

Please consider using an r^2 of 0.001 to ensure the instruments are independent.

From our understanding, the cut-off of r^2 to select SNPs for MR is arbitrary. For example, in the literature suggested by the reviewer where MR was conducted between sleep traits and cardiovascular diseases, PMID 35876003 used $r^2 < 0.6$, PMID 33822910 used $r^2 < 0.1$ and PMID 34315237 used $r^2 < 0.1$. Therefore, our original choice of $r^2 < 0.1$ fits the broadly accepted criteria for choosing SNPs for MR. However, to address this comment, we repeated the linear MR using a stringent cut-off of $r^2 < 0.01$ to select the SNPs. For the analysis between long sleep duration and blood pressure, there were 9 SNPs left for filtering $r^2 < 0.1$ but no SNPs left after filtering $r^2 < 0.01$. Therefore, those 9 SNPs after the filtering of $r^2 < 0.1$ were used for this particular analysis. We then used these selection criteria to re-perform GSMR and these results are presented in the revised manuscript (Figure 8). We have updated the Methods section to reflect this new analysis.

“To identify independent SNP instruments for each exposure (sleep traits), GWAS-significant SNPs ($p_{GWAS} < 5 \times 10^{-8}$) were pruned ($r^2 < 0.01$; linkage disequilibrium (LD), within windows of 10,000 kb). For the MR analysis between long sleep duration and BP, there were no SNPs left after filtering for $r^2 < 0.01$. Therefore, the MR analysis between long sleep duration and BP used 9 SNPs pruned at $r^2 < 0.1$, which has been accepted in conducting MR between sleep traits and cardiovascular diseases.”

Page 20 lines 562 - 567

Whilst including an MR component to verify causal relation is good, the current MR analysis is likely too brief. Including additional sensitivity analyses such as MR-Egger and weighted median would be necessary, as well as reporting strength of instruments (F statistics). Including non-linear MR approach using the doubly ranked stratification method would greatly improve novelty of findings (<https://www.biorxiv.org/content/10.1101/2022.06.28.497930v1>) as some studies suggest the residual method used in earlier non-linear MR studies (not on sleep as this was not assessed) could be biased (PMID: 36528346)

We thank the Reviewer for their insights and recommendations of incorporating these sensitivity analyses and non-linear MR methods in our study. To address this, we conducted additional linear-MR analyses using MR-Egger and weighted median methods. We found that the more stringent r^2 threshold of 0.001 resulted in a lack of power in the MR analyses. Thus, we have elected to repeat the MR-Egger and weighted-MR analyses using a r^2 threshold of 0.01. We also estimated F-stats of instrumental variables (ranging from 34.2 to 51.2 (Table 2), which are above the accepted threshold of 10, demonstrating the validity of the analyses. We have added these details in the methods section. Then we presented the results of GSMR when the results are also significant and in the same direction as at least one of the MR-Egger or weighted median methods (new Figure 8).

We agree with the reviewer that additional non-linear MR analysis adds novelty to our study. Therefore, we also used two methods including the doubly ranked stratification to conduct non-linear Mendelian randomisation between sleep duration and blood pressure. Please see more detailed response below.

One main limitation is possible confounding in observational analyses. I think this is a cross sectional analyses and hence reverse causation could be an issue. Although MR analyses suggested possible associations, the lack of non-linear MR is a main issue, especially when the observational analysis suggested potential non-linearity.

To address the comments from the reviewer, we conducted non-linear MR between sleep duration and blood pressure using two methods, one is PolyMR and the other is SUMnlmr. These methods and the relevant references are included in the manuscript. We focus on sleep duration because this phenotype in linear regression analysis showed non-linear results. Combining results from the two analyses, we found significant non-linear causal associations between sleep duration and systolic blood pressure, although there is a co-existence of non-linear and linear causal relationships.

New text in methods. "We also performed non-linear MR between sleep duration and BP using two methods. We first used PolyMR, which requires raw data and conveniently fits multi-SNP genotype data as instrumental variables. Therefore, SNP genotypes and raw phenotypes from UK Biobank used in the linear regression analyses described above were used. The genotype data of 33 SNPs that were significant in GWAS ($p < 5e-8$) for both sleep duration and BP (presented in the UK Biobank) were extracted and read into PolyMR as instrumental variables. Before MR analysis, BP was adjusted for covariates used in the linear regression analysis and the residual of the outcome variable was fit into PolyMR using default options. The built-in plotting function (`plot_polymr`) was used to visualise the results. Secondly, we performed non-linear MR (doubly-ranked stratification method) using the software implemented in SUMnlmr. As SUMnlmr only accepts one slot for instrumental variables, we used those 33 SNPs to generate polygenic risk score (PRS) using the `plink-score` function for MR analysis in SUMnlmr. As outlined in Burgess et al. (2022), we fitted covariates used in the linear regression analyses into SUMnlmr and performed non-linear MR using default options. Results were plotted following instructions on <https://github.com/amymariemason/SUMnlmr>."

Page 20 lines 578 - 592

Minor comments

Line 151: Please explain the purpose of drawing the correlation plot.

We thank the reviewer for their question. The heat map was intended to be simple way to represent the data generated from the correlation plot. Although the Pearson correlation coefficient values (r) are not strong, the correlation plot was used to give an overview for the direction of the main associations between each sub-category of sleep length, sleep quality and shift work with blood pressure and markers of systemic inflammation. Given that this overview does not provide new data we have elected to

remove it and replace it with a table of the *r* values generated for each correlation (Supplementary Table S11).

Line 158: Township index? Or do you mean Townsend deprivation index?

Yes, we mean Townsend deprivation index. We apologise for any confusion and have corrected this term throughout the main manuscript and the supplementary data file.

Please move the statistical analyses under Mendelian randomisation to statistical analyses

We have now detailed the statistical analyses for the Mendelian randomisation analyses to the “Statistical analyses” subheading.

Page 20 lines 562 – 592

Please insert back the reference group in the tables (e.g. supplemental table 5)

We have edited all results tables to include the reference group in the supplementary materials and data files.

Line 412: What do the authors mean by "have contributed to confounding effects of adjusting for BMI".

Upon our review, the phrasing of this sentence may be the result of an editing mishap. We have reworded the text to “Consistent with the current demographic of shift workers in the workforce, there were relatively fewer female than males undertaking shift work. This may explain why positive associations between day, mixed, night or permanent night shift work with SBP/DBP in males and unstratified datasets were absent in females.”

Page 13 lines 384 - 389

Reviewer #2 (Remarks to the Author):

Thank you for the opportunity to review this interesting paper. The aim of the authors was to assess the associations between sleep length, sleep quality, shift work, inflammatory markers and BP and determine whether these associations differed

across BMI, age, sex categories. The topic is relevant and important given the health and socioeconomic burden of hypertension and sleep problems. Overall, the aims of the study were achieved, and the paper is well written and structured.

We thank the Reviewer for their positive feedback and suggestions, particularly in improving the clarity in our methodology and results content.

Key results

The results are well presented and show that sleep length, sleep quality, and shift work are significantly associated with changes in BP.

Validity

The use of a nationally representative dataset and large sample size makes the findings from this study more robust. However, my suggestion is that the authors make it clear from the beginning of the manuscript that the data was restricted to adults of European white British ancestry; this should appear in the abstract section at the very least, if not the title

We agree this is a major limitation of our study (and the UK Biobank dataset) that should be transparent to readers throughout the full manuscript. The abstract and title have now been edited to include European white British ancestry.

Page 1 Title

Page 2 lines 41 – 42 and 47 - 49

Significance

This study findings support other prior findings showing that problems with sleep, including poor sleep quality, extremes in sleep duration, and working schedules that affect circadian rhythms are associated with adverse health outcomes including increased risk for hypertension. This is especially important given the increasing evidence suggesting that sleep health disparities may be major contributors to other health disparities

Data, Methodology and Analytical approach

The use of multivariable linear regression models to test associations between the sleep parameters, shift work, BP measures were appropriate based on the research objectives of the study. However, I am unable to evaluate the use of GWAS summary based Mendelian randomization (MR) to test for causation since this is beyond the

scope of my expertise.

Just a few comments:

1. Complete the data availability statement “summary data were obtained from....”

Thank you for highlighting this important point, we have now corrected this sentence to “The sleep GWAS summary data were obtained from the Sleep Disorder Knowledge Portal (<http://sleepdisordergenetics.org/>), and the BP GWAS summary data were obtained from the UK Biobank cohort and International Consortium of BP Genome-Wide Association Studies (ICBP).”

Page 21 lines 597 - 600

2. Sleep quality score: was the scoring based on a validated scale/tool or this is a new way of scoring sleep quality developed by the authors? I request the authors to provide reference if the score has been used previously.

In this manuscript, we elected to use this sleep quality score approach, which is used in a number of previous studies also using the UK Biobank cohort including:

- ***“Relationships of sleep traits with prostate cancer risk: A prospective study of 213,999 UK Biobank participants” - Ly et al. (2021)***
- ***“Sleep patterns, genetic susceptibility, and incident cardiovascular disease: a prospective study of 385 292 UK biobank participants” - Fan et al. (2020)***
- ***“Associations Between Sleep Quality and Health Span: A Prospective Cohort Study Based on 328,850 UK Biobank Participants” - Sambou et al. (2021)***
- ***“Baseline Vitamin D Status, Sleep Patterns, and the Risk of Incident Type 2 Diabetes in Data from the UK Biobank Study” - Wang et al. (2020)***

Under the sub-heading “Sleep quality score” in the Methods section, we have edited the sentence “Sleep quality was categorised as either “healthy” (4-5), “moderate” (2-3) or “poor” (0-1) (refer to Supplementary materials), which has been previously used in UK biobank cohort-based studies.”

Page 16 lines 456 - 459

3. *Statistical analyses: AH free model: I have an idea of what AH stands for, but the authors should make that clear for the readers*

We have defined 'AH' as "antihypertensive" in the Results and the Methods sections to avoid confusion for the abbreviation "AH" and provided a short explanation for the AH Free model in the methods.

In the methods: "Antihypertensive (AH) free model', excluding participants reporting the use of antihypertensives in the baseline model".

Page 19 lines 547 - 549

In the Results section:

"Removing participants on antihypertensive medications produced similar results in the 'AH free' and 'AH free BMI adj' models (Supplementary Table S1)."

Pages 5-6 lines 136 - 138

4. *It would be helpful if the authors would outline their justifications for stratification based on age, inflammatory markers, sex, BMI among others. There is a statement in introduction section "previous studies have not evaluated associations stratified for BMI, sex, age or inflammation" but more should be provided.*

We thank the Reviewer for their suggestions and note that it is similar to the first point raised by Reviewer 1 – "Please provide more justification regarding proposed effect modification by sex, age, BMI, and inflammation".

To address these points together, we have edited some of the text in our Introduction and Discussion sections to rationalise our approach of stratifying datasets according to BMI, sex, age and levels of inflammatory markers.

Please see our response to Reviewer 1's comments on this point. In the text: Page 3 lines 74 – 84 and Page 10 – 11 lines 277 – 286.

Results section

The results are well presented and reflect the objectives of the study. I have a few comments below which the authors can consider:

- *The authors should make it clear in the main text which model they are presenting results on. For example, from Lines 224 – 230 the model is not stated*

We have edited the text to state the name of the model throughout the Results section, which is included within the sentences or presented in parentheses.

- *Since a lot of results presented in the main text refer to supplementary tables and figures, I would rather the authors use statistical terms (significant/ not significant) or/and include p-values, rather than using terms such as “strongest”, “modest” when presenting results. This would make it clear to the reader what associations are statistically significant and which are not.*

We have removed descriptive terms such as “strongest” and “modest” and clearly denoted whether outcomes of the regression models presented were significant/not significant as well as including the p-value throughout the Results section.

- *Would the authors clarify this statement in Line 301 “which reflects the strong correlation between BMI and CRP”*

Our data showed that further adjusting for CRP or CRP plus BMI in the baseline model (i.e., the CRP adj and BMI and CRP adj model, respectively) had no effect on primary outcomes of the BMI adj model. These similarities between the BMI adj, CRP adj and BMI and CRP adj models are supported by the moderate correlation between BMI and CRP in the correlation plot (Pearson correlation coefficient = 0.4268).

To detail this, we have reworded the text in the Results to:

“Given that inflammation is associated with increased BP, we investigated the effect of adjusting for levels of CRP alone or in combination with BMI on the associations between sleep length, sleep quality, shift work and SBP/DBP. Our data showed that further adjusting for CRP or CRP plus BMI had no effect on primary outcomes of the BMI adj model, which reflects the strong correlation between BMI and CRP in the correlation plot (Supplementary Table S11 and S12). Poor sleep health (<5hr and >9hr, and poor sleep quality) and night shift work were positively correlated with CRP levels, and leukocyte and neutrophil counts (Supplementary Table S11).”

Page 8 lines 211 - 218

- *Table 2: Improve the readability of this table especially the way p-value are presented.*

To improve the readability of Table 2, we have presented the p-values to only 3 decimal places.

Suggested improvements

No additional analyzes are suggested. For other suggestions/comments, see above

Clarity and context

The findings are clearly presented, and the findings have been discussed well in the context of previous research

References

No obvious problem noted with references, The manuscript references previous literature appropriately.

Your expertise

I am unable to sufficiently evaluate the use of GWAS summary based Mendelian randomization (MR) to test for causation since this is beyond the scope of my expertise.

Reviewer #3 (Remarks to the Author):

Kanki and colleagues demonstrated that sleep duration has a U-shaped association with BP, such that generally, anything on either side of 7h/24 h increases BP. Similar associations exist between sleep quality and night shift work, with permanent shift work and the poorest sleep quality showing a risk for increased BP. Furthermore, there was a U-shape relationship between sleep duration and BP with strongest relationship in people with low CRP. This manuscript must is a lot of hard meticulous work and the authors should be congratulated for these analyses.

We thank the Reviewer for their feedback and suggestions to improve the clarify and interpretation of our findings, particularly in relation to the associations between circadian rhythm-disrupting behaviours and blood pressure.

Some issues are noted here:

Abstract

The authors state that they examine the association between markers of circadian dysregulation, BP and inflammatory markers. There is no examination of circadian dysregulation except for in shift-workers. Therefore, this term should be removed.

Thank you for this suggestion. We have replaced “circadian strain” with “circadian rhythm-disrupting behaviour” throughout the manuscript.

The authors conclude that they have demonstrated an “independent association between circadian rhythm-disrupting behaviours, adverse SBP/DBP regulation and increased inflammatory markers in males and females.” If the association between sleep duration and BP was the strongest in people with CRP levels <1 mg/L, this would indicate that the association is strongest in people with NO inflammation and not increased inflammatory markers.

We agree with the Reviewer’s interpretation of this finding and have re-worded the text in the Abstract accordingly:

“An independent U-shaped relationship was identified between sleep length and SBP/DBP, which was most prominent when inflammatory status was low.” Our concluding sentence in the Abstract also now reads as “This study is the first to demonstrate a positive association between circadian rhythm-disrupting behaviours and SBP/DBP regulation in males and females that was largely independent of age.”

Page 2 lines 41 -41 and lines 47 - 49

Introduction:

The authors introduce the importance of diurnal BP patterns and cardiovascular health. There is no examination of diurnal patterns of BP. The measurements for BP were in duplicate, but it is not known when they were made or the details of how they were obtained. Therefore, it may be prudent to focus the introduction on just office BP. As stated above, in this reviewer’s opinion, other than shift work, there is no evidence for circadian strain because the questions about sleep were over 24h. For instance, the circadian ‘disruption’ is different if a person sleeps from, for e.g., 9 PM to 6 AM vs

someone who sleeps 10:30 PM-6 AM with 1.5h nap in the afternoon when melatonin concentrations are negligible. Therefore, circadian strain cannot be determined. Therefore, in my opinion, the focus should be on sleep and shift work, rather than circadian disruption or strain.

We agree with the Reviewer and appreciate that parameters such as 24-hour rhythms in melatonin or cortisol could not be used to definitively show ‘true’ circadian disruption in the present manuscript (due to availability). Given that office blood pressure is a more accessible and cost-effective approach in monitoring cardiovascular health in large populations, assessing spot measures of blood pressure still makes this study comparable to many others in the current literature.

To avoid any confusion about 24-hour blood pressure, we have removed text related to diurnal patterns in blood pressure and highlighted associations between sleep length, sleep quality and shift work on risks of hypertension and elevated blood pressure.

We have also reworded text in the Methods section under “Study population” to

“While many health and lifestyle factors can be considered circadian rhythm-disrupting behaviours, only ‘sleep length’, ‘sleep quality’ and ‘shift work’ were assessed for simplicity (Supplementary Figure S1). We acknowledge that gold standard markers of circadian disruption such as rhythms of plasma cortisol and melatonin or BP have not been assessed to indicate adverse effects of circadian strain or misalignment on 24-hour BP.”

Page 16 lines 439 - 444

Methods:

Sleep apnea is quite pervasive and people may not know they have it. Simply excluding people with a history of sleep apnea would not exclude people with sleep apnea.

We understand this point the Reviewer has made and acknowledge this is a limitation of our study design. We have edited our Discussion to specifically include this point as a limitation:

“Characterisation of participants into subgroups based on average sleep length and quality, shift schedules and most other covariates were also extracted from self-reported data. We also acknowledge that some participants may experience symptoms but not clinically diagnosed with some health concerns such as sleep apnea or hypertension.”

Page 14 lines 395 - 400

It would be good to state why the 7 h group was considered as standard/ control and not the 8 h group, or both groups combined together.

In our first pass analyses, we considered 8 hours as the reference/control category for the regression analyses. The outcomes of these analyses either did not reveal any significant association or the associations suggested a protective effect of short/long sleep lengths on blood pressure, which is not consistent with the current literature.

Given that the mean SBP and DBP reported were lowest in the 7-hour category, our analyses reveal a U-shaped association between sleep length and blood pressure only after using 7 hours as the reference category. We suspect that because the participants in our analyses are relatively older (mean age of 57 years), 7 hours as the reference category is appropriate, which may be related to age-related declines in sleep length.

A consensus statement from the Consensus Statement of the American Academy of Sleep Medicine and Sleep Research Society has outlined that most studies demonstrate positive associations between sleep durations of less than 6 while data for >8 hours is more heterogeneous in comparison to 7-8 hours in cross-sectional studies. Similarly, greater than 9-hour sleep durations have been positively associated with cardiovascular disease compared to 7-8 hours in cross sectional and prospective studies.

In the “Characteristics of the study population” section under the Results, we have edited the text to include:

“Mean SBP, DBP and BMI were lowest in individuals who reported sleeping an average of 7 hrs/day or in those assigned a healthy sleep score. Thus, these sub-groups were the nominated reference categories for the following linear regression analyses related to each sleep length and sleep quality, respectively.”

Adding 15 and 10 mmHg to people taking BP meds seems too arbitrary. The authors cite one paper who did that, who in turn had cited another paper which state that where this method may be used in clinical trials. That paper also stated that there is virtue in using 10 mm for SBP. The authors should do a sensitivity analysis with both these numbers. Additionally, the removal of participants with HTN meds produced similar, but not identical results with sleep quality. Plus, this was not necessarily a clinical trial.

We thank the Reviewer for this suggestion. We have since come across a systematic review and meta-analyses completed by Marco et al. (2016); doi: [10.1097/MD.0000000000004071](https://doi.org/10.1097/MD.0000000000004071)), which found that “In monotherapy, most drugs achieved 10 to 15 mm Hg SBP and 8 to 10 mm Hg DBP decreases”.

We have repeated the sensitivity analyses using the lower limits as blood pressure corrections (10 mmHg and 8 mmHg for SBP and DBP respectively) in participants who reported the use of anti-hypertensive medications.

A comparison between the original and new correction in the baseline, BMI adj, CRP adj, and BMI & CRP adj models (unstratified datasets) showed that adding 10 and 8 mmHg instead of 15 and 10 mmHg to SBP and DBP, respectively, slightly decreases the beta-estimates. However, it does not largely change the direction of the association or change the status of significance. We have illustrated this as an example for the association between sleep quality and blood pressure in Figure 1 below, and the in the attached spreadsheet.

We have also elected to make a note of this point in our Methods section and state “data not shown”.

“A systemic review and meta-analysis conducted by Paz et al. (2016) found that anti-hypertensive agents reduce SBP by 10-15 mm Hg and DBP by 8-10 mm Hg used provided as a monotherapy. Therefore, we tested the upper and lower limits of these ranges as corrections for the use of anti-hypertensives in this study. BP values for individuals reporting the use of anti-hypertensives were first adjusted by adding 15 and 10 mmHg to SBP and DBP, respectively, as performed in previous studies. A second correction of adding 10 and 8 mmHg to SBP and DBP, respectively, found very similar results to the

first correction (data not shown). We therefore elected to use the first correction (15 and 10 mmHg) as an adjustment in the baseline model.”

Page 17 lines 485 - 493

Figure 1. Sensitivity analyses for corrections applied to systolic and diastolic blood pressure for subjects taking antihypertensive medications. ‘Model1(-BMI)’ refers to the baseline model with no corrections (solid red lines), and ‘Model2(+BMI)’ is the BMI adj model for the association between sleep quality and SBP/DBP (solid blue lines). Adding 10 mmHg and 8 mmHg to SBP and DBP respectively to correct for the use of anti-hypertensive medications is shown by dashed lines and data indicated does not change the outcomes of this association analysis. Data presented are mean beta estimate values and 95% confidence intervals. * $p < 0.05$.

Results:

This title: “Sleep length and sleep quality are associated with a significant change in BP” should perhaps be rephrased as there is no examination of a ‘change’ in BP.

Thank you for this correction. We have edited the text from “Sleep length and sleep quality are associated with a significant change in BP” to “Short/long sleep lengths and unhealthy sleep qualities are associated with BP”.

Page 5 line 121

Because the sample was predominantly white, the conclusions should say so as these may not be generalizable to other groups (as the authors correctly point out).

We agree with the Reviewer that our findings represent the adverse effect of circadian rhythm-disrupting behaviours on blood pressure in white European British individuals.

Although we do not reiterate this in the Results section, we have outlined that Therefore, it is important to repeat these analyses in other ethnicities to reflect the multicultural population in the UK and other developed nations.” in the Discussion. Additionally, we have also edited the Abstract and title of the manuscript to specify our study population in response to comments raised by Reviewer #2.

Page 13 lines 384 - 386

Discussion:

In this reviewer's opinion, any statement about circadian strain should be edited to focus on sleep and shift work. The authors discuss elevation of BP, but there is no change studied; therefore, these statements should be toned down.

We agree with the Reviewer’s comments that we have not addressed changes in blood pressure or risks of hypertension. Therefore, we have removed any text throughout the manuscript alluding to elevations or declines in blood pressure or risk of hypertension when discussing our findings. Instead, we refer to positive and negative associations as appropriate. We have also carefully edited our discussion to strengthen the communication of our primary outcomes and suggest future studies to incorporate data collected over 24 hours to address these very important research question.

The authors should discuss in detail about people with normal CRP levels, i.e., presumably no clinical inflammation, have the strongest relationship between sleep duration and BP.

To address this important point we have included the following new text in our revised manuscript: “Given the links between elevated systemic markers of inflammation and risks of hypertension have been described, we propose that a heightened inflammatory status masks the adverse effects of short or long sleep lengths on BP.”

Add page 13 and line numbers 364-367

It would be educational to the readers to see a scenario translating the beta values to an actual number. For instance, if someone's BP increased by 2 mm because of sleeping for longer or shorter time, even if it may be statistically significant, it may not be clinically meaningful, especially in people who are already taking HTN meds. Taken together the data support an independent link between sleep duration, and BP, but not circadian strain and increased risk of hypertension, especially since this was not a longitudinal study examining people who were normotensive and later became hypertensive.

The beta estimates reported in this study relate to changes in blood pressure across the whole population rather than for an individual. However, a strong linear relationship exists between cardiovascular disease risk and blood pressure for both systolic and diastolic blood pressures. Substantial benefits for patients have been reported even for a reduction of 5mmHg where the risk of CVD and stroke are 21% and 34% lower, respectively. Furthermore, observational studies demonstrate that the cardiovascular risk associated with blood pressure is linear and begins at blood pressure levels within ranges categorized by guidelines as normal or even optimal. Important sex differences in the relative risk of cardiovascular disease associated with hypertension are also apparent. Thus, the outcomes of the present study have important implications for prevention strategies that can modify health risks within a population as well as for individuals.

To highlight this important perspective in our manuscript, we have edited the Discussion section to include:

"These outcomes could have important implications for prevention strategies that can modify health risks within the population as well as for individuals. Previous studies have reported risk reductions of 21% and 34% for cardiovascular disease and stroke respectively, can be achieved in individuals by lowering BP by 5mmHg. Taken together our data suggest that implementation of measures that address lifestyle factors such as shift work, sleep quality and sleep lengths, could have similar cardioprotective benefits even with modest reductions in overall BP."

Page 14 lines 405 - 411

REVIEWER COMMENTS

Reviewer #1 (Remarks to the Author):

This is a study combining both observational and Mendelian randomization design in assessing different sleep traits with hypertension in UK Biobank and GWAS summary statistics. Whilst understanding sleep's effect is of global health importance, this has been explored in previous studies using the same data. There is also room for improvement for the MR part, where addition of non-linear MR would improve overall novelty. Please see below for my comments.

Major comments

Please provide more justification regarding proposed effect modification by sex, age, BMI, and inflammation.

There has been previous studies using the same cohort to address related questions. Please clarify how your study can add to previous literature using the same data apart from sample size difference (PMID: 34869684; 35876003), as well as other Mendelian randomisation studies (PMID: 33822910; 34315237)

There should be more description on the GWAS used, such as covariable adjustments, and the units of the phenotypes. For morning chronotype, what does it mean by 4-level continuous intervals? How does this impact the interpretation of the gene-phenotype estimates?

Please consider using an r^2 of 0.001 to ensure the instruments are independent.

Whilst including an MR component to verify causal relation is good, the current MR analysis is likely too brief. Including additional sensitivity analyses such as MR-Egger and weighted median would be necessary, as well as reporting strength of instruments (F statistics). Including non-linear MR approach using the doubly ranked stratification method would greatly improve novelty of findings (<https://www.biorxiv.org/content/10.1101/2022.06.28.497930v1>) as some studies suggest the residual method used in earlier non-linear MR studies (not on sleep as this was not assessed) could be biased (PMID: 36528346)

One main limitation is possible confounding in observational analyses. I think this is a cross sectional analyses and hence reverse causation could be an issue. Although MR analyses suggested possible

associations, the lack of non-linear MR is a main issue, especially when the observational analysis suggested potential non-linearity.

Minor comments

Line 151: Please explain the purpose of drawing the correlation plot.

Line 158: Township index? Or do you mean Townsend deprivation index?

Please move the statistical analyses under Mendelian randomisation to statistical analyses

Please insert back the reference group in the tables (e.g. supplemental table 5)

Line 412: What do the authors mean by "have contributed to confounding effects of adjusting for BMI".

Reviewer #2 (Remarks to the Author):

Peer review report

Thank you for the opportunity to review this interesting paper. The aim of the authors was to assess the associations between sleep length, sleep quality, shift work, inflammatory markers and BP and determine whether these associations differed across BMI, age, sex categories. The topic is relevant and important given the health and socioeconomic burden of hypertension and sleep problems. Overall, the aims of the study were achieved, and the paper is well written and structured.

Key results

The results are well presented and show that sleep length, sleep quality, and shift work are significantly associated with changes in BP.

Validity

The use of a nationally representative dataset and large sample size makes the findings from this study more robust. However, my suggestion is that the authors make it clear from the beginning of the

manuscript that the data was restricted to adults of European white British ancestry; this should appear in the abstract section at the very least, if not the title

Significance

This study findings support other prior findings showing that problems with sleep, including poor sleep quality, extremes in sleep duration, and working schedules that affect circadian rhythms are associated with adverse health outcomes including increased risk for hypertension. This is especially important given the increasing evidence suggesting that sleep health disparities may be major contributors to other health disparities

Data, Methodology and Analytical approach

The use of multivariable linear regression models to test associations between the sleep parameters, shift work, BP measures were appropriate based on the research objectives of the study. However, I am unable to evaluate the use of GWAS summary based Mendelian randomization (MR) to test for causation since this is beyond the scope of my expertise.

Just a few comments:

1. Complete the data availability statement “summary data were obtained from....”
2. Sleep quality score: was the scoring based on a validated scale/tool or this is a new way of scoring sleep quality developed by the authors? I request the authors to provide reference if the score has been used previously.
3. Statistical analyses: AH free model: I have an idea of what AH stands for, but the authors should make that clear for the readers
4. It would be helpful if the authors would outline their justifications for stratification based on age, inflammatory markers, sex, BMI among others. There is a statement in introduction section “previous studies have not evaluated associations stratified for BMI, sex, age or inflammation” but more should be provided.

Results section

The results are well presented and reflect the objectives of the study. I have a few comments below which the authors can consider:

- The authors should make it clear in the main text which model they are presenting results on. For example, from Lines 224 – 230 the model is not stated
- Since a lot of results presented in the main text refer to supplementary tables and figures, I would rather the authors use statistical terms (significant/ not significant) or/and include p-values, rather than

using terms such as “strongest”, “modest” when presenting results. This would make it clear to the reader what associations are statistically significant and which are not.

- Would the authors clarify this statement in Line 301 “which reflects the strong correlation between BMI and CRP”
- Table 2: Improve the readability of this table especially the way p-value are presented.

Suggested improvements

No additional analyzes are suggested. For other suggestions/comments, see above

Clarity and context

The findings are clearly presented, and the findings have been discussed well in the context of previous research

References

No obvious problem noted with references, The manuscript references previous literature appropriately.

Your expertise

I am unable to sufficiently evaluate the use of GWAS summary based Mendelian randomization (MR) to test for causation since this is beyond the scope of my expertise.

Reviewer #3 (Remarks to the Author):

Kanki and colleagues demonstrated that sleep duration has a U-shaped association with BP, such that generally, anything on either side of 7h/24 h increases BP. Similar associations exist between sleep quality and night shift work, with permanent shift work and the poorest sleep quality showing a risk for increased BP. Furthermore, there was a U-shape relationship between sleep duration and BP with strongest relationship in people with low CRP. This manuscript must is a lot of hard meticulous work and the authors should be congratulated for these analyses.

Some issues are noted here:

Abstract

The authors state that they examine the association between markers of circadian dysregulation, BP and inflammatory markers. There is no examination of circadian dysregulation except for in shift-workers. Therefore, this term should be removed. The authors conclude that they have demonstrated an “independent association between circadian rhythm-disrupting behaviours, adverse SBP/DBP regulation and increased inflammatory markers in males and females.” If the association between sleep duration and BP was the strongest in people with CRP levels <1 mg/L, this would indicate that the association is strongest in people with NO inflammation and not increased inflammatory markers.

Introduction:

The authors introduce the importance of diurnal BP patterns and cardiovascular health. There is no examination of diurnal patterns of BP. The measurements for BP were in duplicate, but it is not known when they were made or the details of how they were obtained. Therefore, it may be prudent to focus the introduction on just office BP. As stated above, in this reviewer’s opinion, other than shift work, there is no evidence for circadian strain because the questions about sleep were over 24h. For instance, the circadian ‘disruption’ is different if a person sleeps from, for e.g., 9 PM to 6 AM vs someone who sleeps 10:30 PM-6 AM with 1.5h nap in the afternoon when melatonin concentrations are negligible. Therefore, circadian strain cannot be determined. Therefore, in my opinion, the focus should be on sleep and shift work, rather than circadian disruption or strain.

Methods:

Sleep apnea is quite pervasive and people may not know they have it. Simply excluding people with a h/o sleep apnea would not exclude people with sleep apnea.

It would be good to state why the 7 h group was considered as standard/ control and not the 8 h group, or both groups combined together.

Adding 15 and 10 mmHg to people taking BP meds seems too arbitrary. The authors cite one paper who did that, who in turn had cited another paper which state that where this method may be used in clinical trials. That paper also stated that there is virtue in using 10 mm for SBP. The authors should do a sensitivity analysis with both these numbers. Additionally, the removal of participants with HTN meds produced similar, but not identical results with sleep quality. Plus, this was not necessarily a clinical trial.

Results:

This title: "Sleep length and sleep quality are associated with a significant change in BP" should perhaps be rephrased as there is no examination of a 'change' in BP.

Because the sample was predominantly white, the conclusions should say so as these may not be generalizable to other groups (as the authors correctly point out).

Discussion:

In this reviewer's opinion, any statement about circadian strain should be edited to focus on sleep and shift work. The authors discuss elevation of BP, but there is no change studied; therefore, these statements should be toned down. The authors should discuss in detail about people with normal CRP levels, i.e., presumably no clinical inflammation, have the strongest relationship between sleep duration and BP. It would be educational to the readers to see a scenario translating the beta values to an actual number. For instance, if someone's BP increased by 2 mm because of sleeping for longer or shorter time, even if it may be statistically significant, it may not be clinically meaningful, especially in people who are already taking HTN meds. Taken together the data support an independent link between sleep duration, and BP, but not circadian strain and increased risk of hypertension, especially since this was not a longitudinal study examining people who were normotensive and later became hypertensive.

REVIEWERS' COMMENTS

Reviewer #1 (Remarks to the Author):

I thank the authors for addressing my comments, and I particularly appreciated the efforts in adding the non-linear MR study. I only have some few minor comments. Well done.

We thank the reviewer for recognising our efforts to complete the additional analyses required to improve the overall novelty of our study.

- Can the authors present the LACE for each stratum in their non-linear MR analyses in a supplemental table? Please also confirm the assumption of these non-linear methods have been checked.

We have now included the estimates of localised average causal effects (LACE) from non-linear MR using SUMnlmr (presented in Supplementary Table S23). We have revised the method section for describing non-linear MR to confirm the assumptions. We have added the following text in our Methods:

“The assumption of the potential existence of non-linear causal relationships was based on the above-described results: 1) significant association analyses between sleep duration and blood pressure and 2) significant linear MR analyses between sleep phenotypes and blood pressure.” “The model assumption of doubly-ranked stratification method which does not require strict parametric assumptions to create strata of the population that have different average levels of exposure is suitable as the exposure of sleep duration is an ordinal variable.”

Line 574-577 page 20 and lines 592- 602 page 21

- Line 247-251: Please consider rewriting this as I think it was confusing as the first sentence says there is non-linearity but the second sentence says there is a mixture of both linearity and non-linearity?

To clarify our interpretation of the causal associations generated from the linear and non-linear MR analyses, we have simplified these sentences to "We observed significant non-linear causal associations between sleep duration and SBP/DBP in the PolyMR32 and SUMnlmr33 methods (Supplementary Figure S10 and S11). Although a significant non-linear causal association was detected, we also note that the association between sleep duration and DBP appeared rather linear, suggesting a linear causal association is favoured between sleep duration and BP."

Lines 245-249 page 9

- Line 235: What is 2 factor MR analysis? [Sorry that I did not notice this in the earlier version]

We apologise for the confusion with our terminology. We have now corrected “2-factor” to “two-sample” throughout the manuscript.

- With increasing demand for transparent reporting, please also consider using STROBE-MR checklist for reporting of the MR analyses.

We thank the reviewer for their suggestion. Our study only used MR to validate many other association results. To address this, we have now added a table (Supplementary information) that contains a more structured breakdown of the MR analyses according to the STROBE-MR checklist.

Reviewer #2 (Remarks to the Author):

This manuscript is well written and study details clearly presented. The authors have adequately addressed the queries and recommendations of the reviewers. I have no major comments and recommend its publication.

We thank the Reviewer for their initial recommendations to improve the clarity in the methods and results, and review of our manuscript.

Reviewer #3 (Remarks to the Author):

The authors have answered my critiques satisfactorily in all the manuscript sections. I do not have further comments.

We thank the reviewer for their earlier suggestions and careful critique of our manuscript.